# Meta Sparse Principal Component Analysis

## Abstract

We study the meta-learning for support recovery (i.e., non-zero entries of the eigenvectors) in high-dimensional Principal Component Analysis. We reduce the sufficient sample complexity in a novel task, with the information that is learned from auxiliary tasks, where a task is defined as a random Principal Component (PC) matrix with its own support. We pool data from all the tasks to execute an improper estimation of a single PC matrix, by maximising the $\ell_1$-regularised predictive covariance. With $m$ tasks for $p$-variate sub-Gaussian random vectors, we establish the sufficient sample complexity for each task to be of the order $O(\sqrt{m^{-1}\log p})$, with high probability. This is very relevant for meta-learning where there are many tasks $m = O(\log p)$, each with very few samples, i.e., $n = O(1)$, in an scenario where multi-task learning fails. For a novel task, we prove that the sufficient sample complexity of successful support recovery can be reduced to $O(\log |J|)$, under an additional constraint that the support of the novel task is a subset of the estimated support union ($J$) from the auxiliary tasks. This reduces the original sample complexity of $O(\log p)$ for learning a single task. Theoretical claims are validated with numerical simulations and the problem of true covariance estimation in brain-imaging and cancer genetics data sets are considered to validate the proposed methodology.

## 1 Introduction

Principal component analysis (PCA) is an important problem in high-dimensional statistical learning with several applications in information theory (Deshpande & Montanari, 2014), image classification (Sun et al., 2019), brain computer interface (Lin et al., 2008), speech recognition (Zheng et al., 2015) among others. Traditionally, PCA focuses on finding the most significant eigenvectors of a covariance matrix calculated from independent and identically distributed (i.i.d.) samples. However, heterogeneous datasets are ubiquitous across various practical fields, and it is reasonable to infer that variables partitioned by categorical attributes such as country, race, gender, and season do not exhibit identical distributions.

Apart from the issue of heterogeneity, high-dimensional PCA brings unique theoretical challenges of its own kind. For instance, it is known that the estimates PCA produces in high-dimensional situations are often inconsistent (Johnstone & Lu, 2009; Nadler, 2008; Paul, 2007). In particular, it has been shown that the eigenvalues of the covariance matrix in high-dimensional data can exhibit unexpected behaviour (Kato, 1949; Bickel & Levina, 2008). Moreover, the eigenvectors of a sample covariance matrix are known to be an inconsistent estimator for the true eigenvectors, i.e., the eigenvectors of the population covariance matrix (Bai et al., 2007). Hence, these challenges present a clear case for carefully understanding conditions under which heterogeneity can be efficiently ameliorated.

In this paper, we refer to the matrix of eigenvectors as PC matrix. Literature about the problem of learning multiple PC matrices is sparse; where there is a compelling argument for efficiently *transferring knowledge already gained from a previous learning task to a new one*, given some structural similarity between previous task and the new PC matrix (referred to henceforth as a "novel" task). This transfer of knowledge is also known as "transfer learning"; Pan & Yang (2010) provide a comprehensive discussion of this topic.

This presents to us three questions. Let $m$ be the number of different PC matrices, referred to henceforth as "auxiliary" tasks, each having $n$ samples of $p$-variate sub-Gaussian random vectors. Then, this paper is mainly concerned with answering the following questions:

1. What is an optimal number of auxiliary tasks $m$ to estimate the underlying PC matrix?

2. Given $m$ auxiliary tasks, what is the optimal number of samples $n$?

3. Given a PC matrix estimated from $m$ tasks, what is the optimal number of samples to estimate the PC matrix of a novel task?

While multi-task learning (Yamane et al., 2016; Yu et al., 2006) allows to partially answer the first two questions (albeit not optimally since it requires to learn every single PC matrix), we focus on meta-learning (Hospedales et al., 2022) which allows to simultaneously and optimally address the above three questions. Furthermore, a common situation in the real world is that there are usually many auxiliary tasks ($m$) but only a small number of samples ($n$) in each task. For instance, for $m = 5000$ taks but only $n = 2$ samples per task, it is infeasible to learn the support of each of the $m$ PC matrices, rendering the multi-task learning method meaningless. In contrast, our meta-learning approach performs "improper" estimation since it estimates a *single* PC matrix with data from different tasks. This enables us to efficiently recover the true support with an optimal number of samples per task.

There are no prior studies on meta-learning for PCA. For other machine learning problems, prior work includes experimental studies, without theoretical guarantees (Lake et al., 2015; Lemke et al., 2015; Vinyals et al., 2016; Ravi & Larochelle, 2017; Finn et al., 2017; Snell et al., 2017; Grant et al., 2018; Yoon et al., 2018; Hospedales et al., 2022). Thus far, theoretical work on meta-learning pertains only to generalisation bounds in learning theory (Maurer, 2005; Pentina & Lampert, 2014; Amit & Meir, 2018; Denevi et al., 2018; Khodak et al., 2019a; Huang et al., 2020; Tripuraneni et al., 2020; Farid & Majumdar, 2021; Chen & Chen, 2022; Guan et al., 2022) and convergence rates in optimisation (Fallah et al., 2020; Finn et al., 2019; Khodak et al., 2019a; Gao & Sener, 2020). In the aforementioned works, performance is only viewed in terms of risks (e.g., misclassification rate, mean squared error) and not in terms of support recovery.

While there has been some work on multi-task PCA (Tiomoko et al., 2023; Yamane et al., 2016), non-asymptotic statistical analyses are not available for multi-task PCA. The closest result for PCA is on structured sparsity (Deng, 2019, Corollary II.5), which sets the sample complexity of multi-task PCA to $n \in O(m \log(mp))$ for each task. The main difference between *multi-task* learning and *meta* learning, is that multi-task learning estimates one PC matrix for each of the different $m$ tasks simultaneously while meta-learning estimates a single PC matrix from different tasks. This allows us to obtain a sample complexity of $n \in O(\sqrt{m^{-1} \log p})$ for each task. For instance, our PCA meta-learning results imply that a regime of many auxiliary tasks $m \in O(\log p)$, only requires few samples $n \in O(1)$ per task. That is, in meta-learning, the number of samples per task ($n$) is constant with respect to the dimension ($p$). Under the same regime, PCA multi-task learning would require $n \in O((\log p)^2)$ samples per task. That is, in multi-task learning, the number of samples per task ($n$) grows poly-logarithmically in the dimension ($p$). Thus, when there are only few samples $n \in O(1)$ per task, multi-task learning fails.

The superiority of meta-learning can be seen across different statistical tasks like regression (Wang & Honorio, 2021), logistic regression (Xie & Honorio, 2022), precision matrix estimation (Zhang et al., 2021), classification (Wang et al., 2021; Wu et al., 2019) and neural networks (Hospedales et al., 2022). Techniques for incorporating aspects of multi-task learning under the meta-learning framework have been studied by Khodak et al. (2019b). Besides our non-asymptotic theoretical results for estimation of PC matrices under sparsity conditions, we also show experimentally that meta-learning produces better estimates than multi-task learning.

## 1.1 Summary of our contributions and organisation of the article

Let $\Sigma^{(1)} \dots \Sigma^{(m)}$ be $m$ many covariance matrices with their corresponding PC matrices $\Pi^{(1)} \dots, \Pi^{(m)}$, both being random perturbations of a true underlying covariance matrix $\Sigma$ with PC matrix $\Pi$. Let $J$ be the support of $\Pi$, i.e., the set of nonzero entries in the eigenvectors. Our key contributions in this paper can be summarised as follows:

(a) The minimax sample complexity for estimating $J$ is given by $n \in O(|J|\sqrt{m^{-1} \log p})$ for each task (Theorems 3.2 and 3.3). Thus, our a meta-learning method works on a challenging scenario where

there are many tasks $m = O(|J|^2 \log p)$, each with very few samples, i.e., $n = O(1)$. In this same scenario, multi-task learning fails.

(b) Given a correctly estimated support union $J$, the optimal sample complexity for estimating a novel task $\Pi^{(m+1)}$ is $O(\log |J|)$, where $|J|$ is the cardinality of the set $J$ (Theorem 4.2). This considerably reduces the original sample complexity of $O(\log p)$ for learning a single task. To the best of our knowledge, this improvement is unique to meta-learning and cannot be replicated by other multi-task learning techniques.

**Outline.** The rest of this paper is outlined as follows: In Section 2, we formally introduce the meta-learning setup and formulate the estimation as a primal-dual problem. In Sections 3 and 4, we formalise the theorems for recovering the sample complexities of meta-learning and learning the novel task. In Section 5, we provide simulation studies to experimentally validate our claims. We end with a summary followed by discussions in Section 6.

## 2 Preliminaries

In this section, we introduce the definitions of the concepts important to our meta-learning problem. A summary of notations used in the paper is illustrated in Table 1.

Table 1: Notations used in the paper

| Notation | Description | Notation | Description |
|---|---|---|---|
| $[a]$ | The first $a$ integers, i.e., $\{1, 2, \ldots, a\}$ | $|J|$ | Cardinality of a set $J$ |
| $A_{i,j}$ | The $i, j^{th}$ cell of matrix $A$ | $U_i$ | The $i^{th}$ coordinate of a vector $u$ |
| $supp(A)$ | Support of $A$, i.e., $\{i : A_{i,i} \neq 0\}$ | $A_{*,i}$ (or $A_{i,*}$) | The $i^{th}$ column (or row) of matrix $A$ |
| $sign(x)$ | The sign of $x, x \in \mathbb{R}$ | $\Sigma$ | The true covariance $A$ |
| $\Sigma^{(i)}$ | Covariance of auxiliary task $i$ | $\lambda_k(A)$ | The $k^{th}$ largest eigenvalue of $A$ |
| $\|u\|_q$ | $l_q$-norm of the vector $u \in \mathbb{R}^p$ | $\|A\|_{q_1, q_2}$ | $\| (\|A_{1,*}\|_{q_1}, \ldots, \|A_{p,*}\|_{q_1}) \|_{q_2}$ |
| $\|A\|_F$ | The Frobenius norm of a matrix A | $\langle A, B \rangle$ | The matrix inner product, i.e., $\text{tr}(A^T B)$ |
| $diag(A)$ | Diagonal of matrix $A$ | $A \otimes B$ | The matrix Kronecker product |
| $\mathbb{D}(A)$ | The diagonal matrix with entries $A_{i,i}$ | $\mathbb{I}_p$ | The $p$-dimensional identity matrix |
| $A_{I,J}$ | Submatrix of $A$ with rows $I$ and columns $J$ | | |

**Multivariate sub-Gaussian Distribution.** This paper is concerned with meta-learning of multiple principal component matrices, each of which are generated randomly. To make inferences about those matrices in sufficient generality, we must first formally define the class of sub-Gaussian vectors with random covariance matrices.

**Definition 2.1.** *For $j \in [n^{(i)}]$ and $i \in [m]$, we say $X_j^{(i)} \in \mathbb{R}^p$ follows a family of random p-dimensional multivariate sub-Gaussian vectors with randomised covariance matrices $\{\Sigma^{(i)}\}$ distribution of size m and parameter $\sigma$ if*

1. *$X_j^{(i)}$ are independent random variables.*

2. *For each $i$ and all $j$, $X_j^{(i)}$ is distributed according to a distribution $P_i$ such that $\mathbb{E}[X_j^{(i)}] = 0$, $\text{Cov}\left(X_j^{(i)}\right) = \Sigma^{(i)}$.*

3. *$\frac{X_{l,j}^{(i)}}{\Sigma_{l,l}^{(i)}}$ is sub-Gaussian with parameter $\sigma$, $\forall i \in [m]$ and $j \in [n^{(i)}]$ and $l \in [p]$.*

**Remark 2.2.** *We would like to point out that such definitions are standard in the literature and have been used extensively in prior work (see for instance Zhang et al. (2021)) for modelling sub-Gaussian random variables with randomised covariance matrices under the meta-learning setup. We have deferred the definition of sub-Gaussian and sub-exponential distributions to Appendix A.*

**Problem Formulation.** In this paper, we have focused on estimating the support of the PC matrix of a multivariate sub-Gaussian distribution. We first estimate a superset of the support of the PC matrices. Then we solve a novel task utilising the recovered support union.

To be precise, let for $i \in [m]$, $\Sigma$ be the true underlying covariance matrix and $\Sigma^{(i)}$ be perturbations of $\Sigma$, with each $\Sigma^{(i)}$ forming the covariance matrix of an auxiliary task $i$. Then, we are interested in using those auxiliary tasks to recover support union $J$ of the PC matrix $\Pi$ corresponding to $\Sigma$, i.e., $J = supp(\Pi)$. This is because we are interested in a novel task where the support of its PC matrix is assumed to be a subset of the support of $\Pi$. This allows us to achieve a substantial reduction in sample complexity from $O(\log(p))$ to $O(\log |J|)$ while estimating the support in the novel task.

## 2.1 A Meta-Learning Model

In this section, we define a generative model for the population covariance matrices. Let $\Sigma$ be a positive semi-definite, real symmetric matrix. For some orthogonal matrix $U$, and positive diagonal matrix $\Lambda$, the spectral decomposition of $\Sigma$ gives us,

$$\Sigma = U\Lambda U^T := \sum_{l=1}^{p} \lambda_l(\Sigma)U_{*,l}U_{*,l}^T, \tag{2.1}$$

where $U_{*,l}$ are orthonormal vectors and $\lambda_i$ are the diagonal entries of $\Lambda$. If $\lambda_l(\Sigma) > \lambda_{l+1}(\Sigma)$, $\forall l$, then,

$$\Pi = \sum_{l=1}^{k} U_{*,l}U_{*,l}^T \tag{2.2}$$

gives us the unique $k$-component PC matrix of $\Sigma$. With random matrices $R^{(i)}$ and $D^{(i)}$, we generate a sequence of random matrices $\Sigma^{(1)}, \ldots, \Sigma^{(m)} \in \mathbb{R}^p$. For each $i \in [m]$, the covariance matrix $\Sigma^{(i)}$ corresponding to the $i$-th auxiliary task is generated as,

$$\Sigma^{(i)} = R^{(i)}U(\Lambda + D^{(i)})U^T \left(R^{(i)}\right)^T. \tag{2.3}$$

Referring to Definition 2.1, we assume that for $j \in [n^{(i)}]$, our data, $X_j^{(i)}$ is a random $p$-dimensional multivariate sub-Gaussian random variable with randomised covariance matrix $\Sigma^{(i)}$. Let $S^{(i)} := \frac{1}{n^{(i)}} \sum_{j=1}^{n^{(i)}} X_j^{(i)}(X_j^{(i)})^T$ be the sample covariance matrix corresponding to the auxiliary task $\Sigma^{(i)}$. A "pooled" estimator $S$ of $\Sigma$ is defined below:

$$S := \frac{1}{m} \sum_{i=1}^{m} S^{(i)}. \tag{2.4}$$

## 2.2 The Predictive Covariance Loss Function

In this subsection, we describe the objective function that we maximise to obtain the improper estimator. Recall from Theorem 1 in Overton & Womersley (1992) that PCA can be interpreted as a covariance maximisation technique maximising the predictive covariance $\operatorname{tr}(\operatorname{Cov}(X, HX|H))$. Observe that,

$$\operatorname{tr}\left(\operatorname{Cov}\left(X, HX|H\right)\right) = \langle \Sigma, H \rangle,$$

for $\operatorname{Cov}(X) = \Sigma$ and a rank $k$ orthogonal matrix $H$. Accordingly, the $k$-component sparse PCA seeks the rank-$k$ orthogonal projection matrix $H$ such that for a penalty $\rho > 0$ the penalised predictive covariance

$$\langle \Sigma, H \rangle - \rho\|H\|_{1,1} \tag{2.5}$$

is maximised. Crucial to getting a consistent estimate of the principal component matrix is its identifiability. Unless the PC matrix is unique, there is no hope to correctly estimate the PC matrix. For instance, if $\Sigma = \mathbb{I}_p$, then every $k$-dimensional subspace of $\mathbb{R}^p$ is a valid principal component. Slightly abusing notation, by $\lambda_k$ we denote $\lambda_k(\Sigma)$ and by $\lambda_{k+1}$ we denote $\lambda_{k+1}(\Sigma)$. We make our first assumption about the uniqueness of the PC matrix, denoted in terms of the spectral gap of the covariance matrix $\Sigma$.

**Assumption 1** (Uniqueness). *Let $\lambda_k$ and $\lambda_{k+1}$ be the $k$-th and $k+1$-th eigenvalues of $\Sigma$. Then, $\lambda_k - \lambda_{k+1} > 0$.*

**Remark 2.3.** *The above condition ensures that the principal subspace is identifiable. Subsequently, we need to define the notion of sparsity of the PC matrix. Recall from Equation (2.2) that $\Pi$ is a positive semi-definite matrix. Consequently, the $i$-th row/column of $\Pi$ is 0 is non-zero if and only if $\Pi_{i,i} \neq 0$. Hence, it is sufficient to recover the set of non-zero entries in the diagonal of $\Pi$. Hence, we can make our sparsity assumption.*

**Assumption 2** (Sparsity). *Let $\Pi$ be the $k$-component PC matrix corresponding to $\Sigma$. The support union $J = supp(\Pi)$ fulfills $|J| \ll p$.*

Throughout Section 3, we will assume that $\Sigma$ satisfies both Assumption 1 and Assumption 2.

### 2.3 The Improper Objective Function

As illustrated in Section 2, we recover the support union by estimating the true covariance matrix. To be specific, we pool together all the samples from $m$ tasks and maximise the $l_1$-regularised predictive covariance between the true and the estimated covariance matrices, i.e., we solve the following optimisation problem with regularisation constant $\rho > 0$

$$\hat{\Pi} := \arg\sup_H \sum_{i=1}^m T_i \left\{ \langle S^{(i)}, H \rangle - \rho \|H\|_{1,1} \right\} \tag{2.6}$$

$$\text{subject to } H \in \mathcal{F}^k, \text{ where,}$$

$$\mathcal{F}^k := \{ M \in \mathbb{R}^{p \times p} : 0 \preceq M \preceq \mathbb{I}_p \text{ and } \operatorname{tr}(M) = k \},$$

and $T_i$'s are weights proportional to the number of samples $n^{(i)}$ for each $i$. For clarity of exposition, we assume that $n^{(i)} = n$ for all values of $i$ for the reminder of this paper. This allows us to assume $T^{(i)} = \frac{1}{m}$ without losing generality. Consequently, we can rewrite the objective function in (2.6) as

$$\hat{\pi} = \arg\sup_H \langle S, H \rangle - \rho \|H\|_{1,1} \text{ subject to } H \in \mathcal{F}^k, \tag{2.7}$$

where $S$ is as defined in Equation (2.4). It is noteworthy that Equation (2.6) is an improper estimation technique since it estimates a single PC matrix with data from different tasks. This enables us to efficiently recover the true support with an optimal number of sample size per task (see Section 3).

For the next step, suppose that we have successfully recovered the true support union $J$ in the first step. Then for the novel task, $(m+1$-th task), we can assume that the support of its PC matrix is also a subset of the recovered support union. Then, for a noisy estimate $S^{(m+1)}$ of $\Sigma^{(m+1)}$ we propose the following $l_1$-regularised predictive covariance loss function

$$\langle S^{(m+1)}, H \rangle - \rho \|H\|_{1,1} \text{ subject to } H \in \mathcal{F}^k$$
$$\text{and } supp(H) \subseteq J. \tag{2.8}$$

Due to presence of the extra constraint, the above estimation technique reduces the dimension of the optimisation problem from $p \times p$ to $|J| \times |J|$.

## 3 Support Union Recovery for the Auxiliary Tasks

In this section, we formally show that a sample complexity of $O(\sqrt{m^{-1} \log p})$ for each task, is sufficient for the recovery of the support union via optimising Equation (2.7). Thus, our a meta-learning method works on a challenging scenario where there are many tasks $m = O(\log p)$, each with very few samples, i.e., $n = O(1)$. In this same scenario, multi-task learning fails. One key task in achieving the said sample complexity is to carefully upper bound the tail probabilities of $\|S - \Sigma\|_{\infty,\infty}$. To that end, we prove the following tail probability for a family of multivariate sub-Gaussian vectors with randomised covariance matrices. Let $S^{(i)} = \frac{1}{n} \sum_{t=1}^n X_t^{(i)} (X_t^{(i)})^{\mathrm{T}}$.

**Lemma 3.1.** *Let $\Sigma^{(i)}$ be a sequence of random covariance matrices such that $\left\|\Sigma^{(i)}\right\|_{\infty,\infty} \leq \eta$. Then, for $\{X_j^{(i)}\}_{1\leq j\leq n, 1\leq i\leq m}$ following a family of random p-dimensional multivariate sub-Gaussian distributions with parameter $\sigma$ and random covariance matrices $\Sigma^{(i)}$,*

$$\mathbb{P}\left(\left\|\frac{1}{m}\sum\left(S^{(i)}-\Sigma^{(i)}\right)\right\|_{\infty,\infty} \geq \varepsilon\right) \leq \frac{p(p+1)}{2}e^{-\frac{nm\varepsilon^2}{512\sigma^4\eta}},$$

*whenever $0 < \varepsilon < 32\eta\sigma^2$.*

The proof of this lemma is presented in Appendix B.1. Recall from Equation (2.3) that we defined each auxiliary task $\Sigma^{(i)}$ in terms of random matrices $R^{(i)}$ and random matrices $D^{(i)}$. Key to the process of improper estimation is the control of the maximal loss between the auxiliary tasks $\Sigma^{(i)}$ and the true underlying task $\Sigma$. In other words, we need to find theoretical guarantees for $\left\|\Sigma^{(i)}-\Sigma\right\|_{\infty,\infty}$. We achieve it by controlling the variance and support of the perturbations $R^{(i)}$ and $D^{(i)}$. To that end, we make the following assumption.

**Assumption 3.** *Assume that for each $1 \leq i \leq m$, $R^{(i)}$ and $D^{(i)}$ in Equation (2.3) are random matrices that fulfill the following conditions:*

1. *Each $R^{(i)}$ is i.i.d. with,*

$$\mathbb{E}\left[R^{(i)} \otimes R^{(i)}\right] = \mathbb{I}_{p^2} \tag{3.1}$$

   *Furthermore, we assume that*

$$\|R^{(i)} - \mathbb{I}_p\|_{\infty,1} \leq c\sqrt{g_p} - 1 =: \mathbb{C}_R \tag{3.2}$$

   *almost surely for some $g_p$ bounded from above by $2p$, and $c$ is a positive universal constant. Without losing generality, we put $c = 1$. The proofs would proceed very similarly in all other cases.*

2. *Each $D^{(i)}$ is i.i.d. with $\mathbb{E}[D^{(i)}] = 0$. Moreover, for some constant $L > 0$,*

$$\lambda_1(D^{(i)}) < L \text{ almost surely.} \tag{3.3}$$

White noise additive models have been used extensively in prior literature to study precision and covariance matrices (Wang & Honorio, 2021; Zhang et al., 2021). It is noteworthy that under our assumptions, we can recover additive models from Equation (2.3) when $R^{(i)} = \mathbb{I}_p$ for every $i$. However, $R^{(i)}$ suffers from curse of dimensionality, and consequently Equation (3.2) is required to control the rate of growth of $R^{(i)}$. We note that Assumption 3.2 is weak, but still sufficient to produce a minimax optimal sample complexity. However, it is not a necessary assumption. In this light, a necessary condition on the matrices $R^{(i)}$ is by itself an interesting question and is part of a future work. Our final technical assumption is given below.

**Assumption 4.** *If $\Sigma$ is a covariance satisfying Assumption 2. Then,*

$$\frac{8|J|}{\lambda_k - \lambda_{k+1}}\|\Sigma_{J^c,J}\|_{2,\infty} =: (1-\alpha) < 1.$$

This condition previously appears in Lei & Vu (2015) and recovers the conditions by Amini & Wainwright (2008) when $\Sigma_{J^c,J} = 0$. It is also similar to the irrepresentability condition in the seminal papers (Zhao & Yu, 2006; Meinshausen & Bühlmann, 2006; Meinshausen & Yu, 2009) for $l_1$ penalised sparse regression and lasso type recovery. As far as we know, this is the most general condition for the support recovery of PC matrices and encompasses the conditions when $\Sigma$ is block diagonal. For ease of exposition, we define the following notation:

$$\lambda_{diff} := \lambda_k - \lambda_{k+1} \qquad \lambda^\dagger := 2\left(\mathbb{C}_R + 1\right)^2\left(\lambda_1(\Sigma) + L\right),$$
$$\rho_1 := 4\lambda^\dagger\sqrt{\frac{\log(p)}{m}} \qquad \rho_2 := 16\sqrt{2\sigma^4\lambda^\dagger\frac{\log(p+1)}{nm}}.$$

The following theorem establishes the $O(|J|^2\log p/m)$ per-task sample complexity for the support union recovery from multiple auxiliary tasks.

**Theorem 3.2** (Support Union Recovery)**.** *Let Assumption 3 and Assumption 4 hold. Then there exists a large enough constant $\mathbb{C}_\pi > 0$ such that, whenever*

$$\alpha \max\{\rho_1, \rho_2\} < \rho, \text{ and}$$

$$\rho < \min\left\{ \frac{(\lambda_{diff})\min_{j\in J}\Pi_{j,j}}{16|J|}, \frac{(\lambda_{diff})^2}{4|J|\left(\lambda_{diff} + 8\lambda_1(\Sigma)\right)} \right\}$$

*then, with probability at least $1 - \frac{2}{p^2} - \frac{1}{2(p+1)^2}$ the solution $\hat{\Pi}$ to the objective function Equation (2.7) satisfies,*

    1. $supp(\hat{\Pi}) = J,$                         2. $\left\|\hat{\Pi} - \Pi\right\|_{\infty,\infty} \le \mathbb{C}_\pi \rho.$

In other words, Theorem 3.2 gives us our required sample complexity for sign consistency and error bound. Recalling $\mathbb{C}_R$ from Equation (3.2) we find that $\lambda^\dagger = (\lambda(\Sigma) + L) \times g_p$. Therefore, as long as

$$\alpha \max\left\{ 4\lambda^\dagger\sqrt{\frac{\log(p)}{m}}, 16\sqrt{2\sigma^4\lambda^\dagger\frac{\log(p+1)}{nm}} \right\} < \min\left\{ \frac{(\lambda_{diff})\min_{j\in J}\Pi_{j,j}}{16|J|}, \frac{(\lambda_{diff})^2}{4|J|\left(\lambda_{diff} + 8\lambda_1(\Sigma)\right)} \right\}$$

with probability at least $1 - \frac{2}{p^2} - \frac{1}{2(p+1)^2}$, we recover the true support. Our next objective is to provide a matching lower bound for the previous per-task sample complexity.

**Theorem 3.3.** *In the setup of Theorem 3.2, there exists an estimation problem such that, for all $p$ large enough any estimator $\hat{J}$ of the support union $J$, if $n \le \frac{g_p|J|\log p - |J|\log|J|}{2m}$., then*

$$\mathbb{P}\left(\hat{J} \ne J\right) \ge \frac{1}{4}.$$

The proof of the above theorem is provided in Appendix B.3.

**Remark 3.4.** *Observe that, by setting $n = 1$, we also obtain a corresponding error lower bound whenever $m \le \frac{g_p|J|\log|p| - |J|\log|J|}{2}$ which matches the upper bound up to a factor of $g_p$.*

### 3.1 Sketch of Proof for Theorem 3.2

The complete proof of this theorem may be found at Appendix B.2 and we only make the outline here. The proof of Theorem 3.2 is via primal-dual witness method. It starts by writing down the objective function Equation (2.7).

$$\max_{H \in \mathcal{F}^k} \langle S, H \rangle - \rho\|H\|_{1,1}.$$

Define the subspace

$$\mathbb{B}_p := \left\{ Z \in \mathbb{R}^{p \times p} : diag(Z) = 0, Z = Z^T, \|Z\|_{\infty,\infty} \le 1 \right\}.$$

Using strong-duality of $\|\cdot\|_{1,1}$, we can rewrite the objective function as,

$$\max_{\substack{H\in\mathcal{F}^k \\ Z\in\mathbb{B}}} \langle S, H\rangle - \rho\langle H, Z + \mathbb{I}_p\rangle \equiv \max_{\substack{H\in\mathcal{F}^k \\ Z\in\mathbb{B}}} \langle S, H\rangle - \rho\langle H, Z\rangle - \rho k$$

$$\equiv \max_{\substack{H\in\mathcal{F}^k \\ Z\in\mathbb{B}}} \langle H, S - \rho Z\rangle, \tag{3.4}$$

where the last step follows from the fact that $k$ is a constant. To maximise Equation (3.4), we need to find solutions $(\hat{H}, \hat{Z})$ such that Karush-Kuhn-Tucker (KKT) (Boyd et al., 2004) conditions

$$\hat{Z}_{i,j} = sign(\hat{H}_{i,j}), \forall i \ne j \text{ and } \hat{H}_{i,j} \ne 0, \tag{3.5}$$

$$\hat{Z}_{i,j} \in [-1, 1], \forall i \ne j \text{ and } \hat{H}_{i,j} = 0, \tag{3.6}$$

$$\hat{H} = \arg\max\langle S - \rho\hat{Z}, H\rangle, \tag{3.7}$$

are satisfied. Proceeding according to the primal-dual witness method, we construct the constrained optimisation problem where the $supp(H) \subseteq J$.

$$\max_{H \in \mathcal{F}^k, supp(H) \subseteq J} \langle S, H \rangle - \rho \|H\|_{1,1}. \tag{3.8}$$

Let $(\tilde{H}, \tilde{Z})$ be the corresponding primal and dual variables. The proof selects an appropriate $|J| \times |J|$ orthonormal matrix $Q$ according to Lemma A.6 with the correct rotation such that the Frobenius norm between the true eigenvectors $U_{J,*}$ and the estimated eigenvectors $\hat{U}_{J,*}$ is small. Then we propose the following solution $(\hat{H}, \hat{Z})$ to the dual objective Equation (3.4). It is noteworthy that this construction is similar to that in Theorem 1 in Lei & Vu (2015).

$$\hat{H} = \tilde{H},$$
$$\hat{Z}_{J,J} = \tilde{Z}_{J,J}, \tag{3.9}$$
$$\hat{Z}_{ij} = \frac{1}{\rho}\big\{S_{ij} - \langle Q_{i,*}, \Sigma_{J,j}\rangle\big\}, (i,j) \in J \times J^c, \tag{3.10}$$
$$\hat{Z}_{ij} = \frac{1}{\rho}(S - \Sigma)_{ij}, (i,j) \in J^c \times J^c, i \neq j. \tag{3.11}$$

The rest of the proof follows in 4 parts.

1. Finding an appropriate $Q$ matrix and an appropriate tail probability bound under Assumption 3 by using a combination of Matrix Chernoff bound (see Appendix A) and Lemma 3.1.

2. Proving that the proposed solution satisfies the KKT conditions (3.5), (3.6), and (3.7) with high probability for our choices of $\rho$ and is unique.

3. Find an upper bound for estimation error. We achieve this through the following upper bound, $\left\|\hat{H} - \Pi\right\|_{\infty,\infty} < 2\frac{\rho|J|}{\lambda_{diff}}$, which is a mild variation of a similar upper bound from Lemma 2 in Vu et al. (2013). Thus, for sufficiently small enough $\rho$, we prove that $sign(\hat{\Pi}) = sign(\Pi)$, and our estimate is sign consistent.

4. Showing that the proposed solution is sign consistent using a similar technique to part 3.

## 4 Support Recovery for the Novel Task

In this section, we show that $O(\log|J|)$ sample complexity is sufficient for the support recovery for a novel task where $J$ is the recovered support union in Section 3. This considerably reduces the original sample complexity of $O(\log p)$ for learning a single task. We also show that for the novel task, the estimation problem can be reduced from estimating a $p \times p$ dimensional matrix to a $|J| \times |J|$ dimensional matrix. Let $n^{(m+1)}$ be the number of samples in the novel task. We begin by establishing the reduction in dimension. Let $\Sigma^{(m+1)}$ be the variance matrix corresponding to the novel task with spectral decomposition

$$U^{(m+1)}\Lambda^{(m+1)}\left(U^{(m+1)}\right)^T,$$

and the $k$-component PC matrix $\Pi^{(m+1)}$. Throughout the rest of this section, we assume that $\Sigma^{(m+1)}$ satisfies Assumption 1 and Assumption 2. We write $\Sigma^{(m+1)}$ and its estimator $S^{(m+1)}$ as the following block matrices

$$\Sigma^{(m+1)} = \begin{bmatrix} \Sigma_{J,J}^{(m+1)} & \Sigma_{J,J^c}^{(m+1)} \\ \Sigma_{J^c,J}^{(m+1)} & \Sigma_{J^c,J^c}^{(m+1)} \end{bmatrix} \quad S^{(m+1)} = \begin{bmatrix} S_{J,J}^{(m+1)} & S_{J,J^c}^{(m+1)} \\ S_{J^c,J}^{(m+1)} & S_{J^c,J^c}^{(m+1)} \end{bmatrix}. \tag{4.1}$$

We obtain the $k$-component PC matrix of the novel task $\Pi^{(m+1)}$ by optimising the objective function,

$$\max_{H \in \mathcal{F}^k} \langle S^{(m+1)}, H \rangle - \rho \|H\|_{1,1}. \tag{4.2}$$

However, we have already recovered the support union $J$. Hence, we can add another constraint on $H$: $supp(H) = J$. Moreover, since the row and column $i$ of the principal component $\Pi^{(m+1)}$ is 0 if and on $\Pi_{i,i}^{(m+1)} = 0$, we can rewrite this additional constraint as,

$$H = \begin{bmatrix} H_{J,J} & 0 \\ 0 & 0 \end{bmatrix}.$$

Since $H \in \mathcal{F}^k$, the maximisation problem only makes sense if $|J| > k$. This also automatically implies that $H \in \mathcal{F}_J^k$, where $\mathcal{F}_J^k := \left\{ M \in \mathbb{R}^{|J| \times |J|} : 0 \preceq M \preceq \mathbb{I}_{|J|} \text{ and } \operatorname{tr}(M) = k \right\}$. Therefore, the objective corresponding to the novel task maximises $\langle S^{(m+1)}, H \rangle - \rho\|H\|_{1,1}$ subject to $H \in \mathcal{F}^k$ and $supp(H) \subseteq J$. This is equivalent to maximising

$$\hat{\Pi}^{(m+1)} := \arg \sup_{H_{J,J}} \ \langle S_{J,J}^{(m+1)}, H_{J,J} \rangle - \rho\|H_{J,J}\|_{1,1}, \text{ for}$$

$$H_{J,J} \in \mathcal{F}_J^k \tag{4.3}$$

where $supp(\hat{\Pi}) = J$ from Theorem 3.2 is the recovered support union, and $H_{J,J}, S_{J,J} \in \mathbb{R}^{|J| \times |J|}$. The following proposition on the eigenvalues and eigenvectors of $\Sigma_{J,J}^{(m+1)}$ shows that the estimation problem can be reduced from estimating a $p \times p$ dimensional matrix to a $|J| \times |J|$ dimensional matrix.

**Proposition 4.1.** *For $1 \le l \le k$, $U_{l,J}^{(m+1)}$ are the first $k$ eigenvectors of $\Sigma_{J,J}^{(m+1)}$ with corresponding eigenvalues $\lambda_l^{(m+1)}$.*

The proof of this proposition is given in Appendix B.4. Following from the fact that $\Sigma^{(m)}$ satisfies Assumption 1 and Assumption 2, Proposition 4.1 establishes as an immediate consequence that

$$\lambda_k^{(m+1)} - \lambda_{k+1}^{(m+1)} > 0, \text{ and, } J^{(m+1)} = supp(\Pi^{(m+1)}).$$

This establishes that the $k$-component PC matrix of $\Sigma_{J,J}^{(m+1)}$ is sparse and unique, as well as the solution to a lower dimensional objective function.

Overloading notation, define $\lambda_{diff}^{(m+1)} := \lambda_k^{(m+1)} - \lambda_{k+1}^{(m+1)}$. We make the following technical assumption on the entries of $\Sigma_{J,J}^{(m+1)}$.

**Assumption 5.** *Let $\Sigma^{(m+1)}$ be the covariance matrix corresponding to the novel task and $J$ be the recovered support union. Then,*

$$\frac{8|J^{(m+1)}|}{\lambda_{diff}^{(m+1)}} \|\Sigma_{(J^{(m+1)})^c, J^{(m+1)}}^{(m+1)}\|_{2,\infty} =: \left(1 - \alpha^{(m+1)}\right) < 1.$$

This condition is similar to Assumption 4, and as we mention before, previously appears in (Lei & Vu, 2015) and recovers the conditions by Amini & Wainwright (2008) when $\Sigma_{J^c,J} = 0$. With this, we can now proceed to state our theorem regarding support recovery in novel task. We first define

$$\mathbb{C}_{n1} := \frac{1}{32\lambda_1\left(\Sigma^{(m+1)}\sigma^2\right)},$$

$$\mathbb{C}_{n2} := \frac{4|J^{(m+1)}|\left(1 + \frac{8\lambda_1^{(m+1)}}{\lambda_k^{(m+1)} - \lambda_{k+1}^{(m+1)}}\right)}{\lambda_k^{(m+1)} - \lambda_{k+1}^{(m+1)}},$$

$$\mathbb{C}_{n3} := \frac{\left(\lambda_k^{(m+1)} - \lambda_{k+1}^{(m+1)}\right) \min_{j \in \left[|J^{(m+1)}|\right]} \Pi_{j,j}^{(m+1)}}{4|J^{(m+1)}|}.$$

The following theorem shows the $O(\log|J|)$ sample complexity for support recovery of the novel task.

**Theorem 4.2** (Novel Task Support Recovery). *Let $\Sigma^{(m+1)}$ be the covariance matrix corresponding to the novel task satisfying Assumption 5 and $J$ be the recovered support union. Furthermore, assume that,*

$$n^{\frac{1}{3}} > \max\{\mathbb{C}_{n1}, \mathbb{C}_{n2}, \mathbb{C}_{n3}\} \ and,$$

$$\frac{1}{n^{1/3}\alpha^{(m+1)}} < \rho < \frac{\lambda_k^{(m+1)} - \lambda_{k+1}^{(m+1)}}{4|J^{(m+1)}|\left(1 + \frac{8\lambda_1^{(m+1)}}{\lambda_k^{(m+1)} - \lambda_{k+1}^{(m+1)}}\right)},$$

*then, with probability at least $1 - \frac{|J|(|J|+1)}{2}e^{-\frac{n^{1/3}}{8\mathbb{C}\sigma^2\left(2\lambda_1(\Sigma^{(m+1)})\right)}}$, the maximiser $\hat{\Pi}^{(m+1)}$ to the objective function Equation (2.8) satisfies, $supp(\hat{\Pi}^{(m+1)}) = J^{(m+1)}$.*

The proof of this theorem is deferred to Appendix B.5.

## 5 Experiments

In this section, we empirically validate our theoretical claims in Sections 3 and 4.

**Synthetic Experiments.** For all the experiments in this subsection, we let $p = 50$, $k = 5$ and $|J| = 5$ and perform 100 repetitions of each setting. We first consider the setting of different sample sizes. We choose $n \in \{3, 5, 7, 9\}$ and use $\rho = \sqrt{\frac{\log(p+1)}{mn}}$ for all pairs of $(m, n)$. The number of tasks $m$ is rescaled to $T$ defined by $\frac{mn}{\log(p+1)}$. For each $j \in [n]$, $X_j^{(i)}$ is generated from a Gaussian distribution with mean a $p$-dimensional 0 vector and covariance matrix $\Sigma^{(i)}$. Figure 1 depicts the outcome of our experiments. For different choices of $n$, the graphs overlap each other perfectly (both $\mathbb{P}(\hat{J} = J)$ and $\|\hat{\Pi} - \Pi\|_{\infty,\infty}$). Then, we took the recovered support union and derived the probability of support recovery for a novel task. We perform the experiments when there are $2, 3$, or $4$ extra zeros and all of those have greater than $95\%$ probability of accurately identifying the extra zeros.

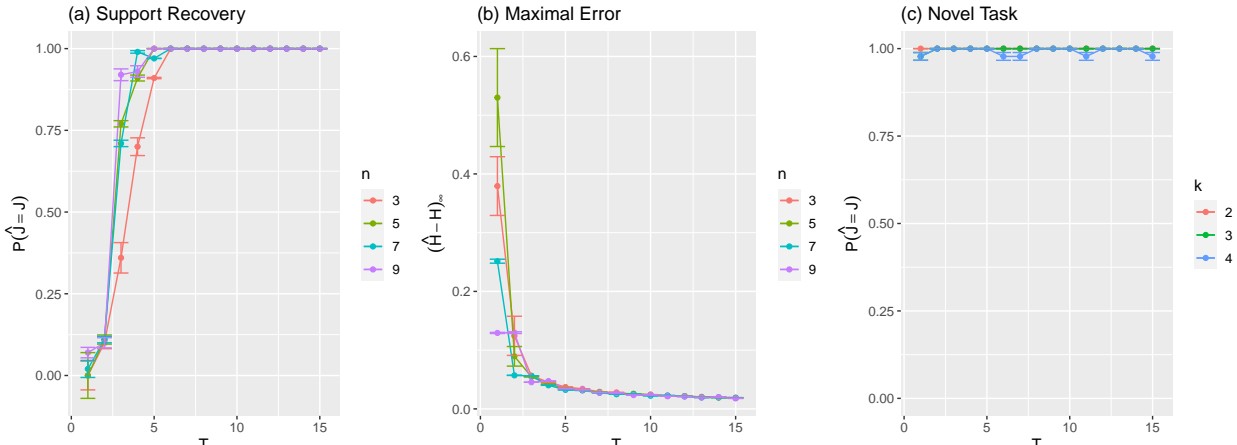

Figure 1: (a), (b): Simulations for Theorem 3.2 on the probability of support union recovery and maximal error under various settings of $n$. $\rho = \sqrt{\frac{\log(p+1)}{mn}}$. The x-axis is set by $T := \frac{mn}{\log(p+1)}$ varying from $\{1, \ldots, 15\}$ for support union recovery. (c): Probability of support recovery for novel task. $\rho = \sqrt{\frac{\log(|J|+1)}{n}}$. The x-axis is set by $T := \frac{n}{\log(|J|+1)}$ varying from $\{10, \ldots, 25\}$

In Appendix C.1 we provide more details and show extra results for the uniform distribution as well as a mixture of sub-Gaussian distributions. Furthermore, we conduct comparisons to show the superiority of meta-learning over multi-task learning Appendix C.2 and over single task learning in Appendix C.3.

**Real World Experiments.** In Appendix C.4, we show that our method obtains anywhere between $85 - 95\%$ accuracy in a real world brain-imaging dataset and similar results in a cancer dataset. We also found consistent results with brain imaging (Rubia et al., 2000; Yan et al., 2005; Baron-Cohen et al., 2006) and cancer genetics (Huang et al., 2014; Sun et al., 2020; Feber et al., 2004; Hu et al., 2009) studies. These findings demonstrate the applicability and interpretability of meta-learning in practical applications.

## 6 Concluding Remarks

We believe we are taking a meaningful and important first step that will potentially motivate further developments in the community. Dependencies in the data, such as when the data is from a time series makes an interesting scenario. Similarly, it is also unclear how estimation should proceed when there is missing data. Finally, we would like to point out that extending our results to kernel methods would also be an interesting future endeavor.

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

## A  Technical Desiderata

In this section of the appendix, we state some technical definitions and lemmas. We begin with the definition of a real valued sub-Gaussian random variable (Zhang et al., 2021).

**Definition A.1.** *A random variable $X$ is called sub-Gaussian with parameter $\sigma \geq 0$ if, $\forall \rho \in \mathbb{R}$*

$$\mathbb{E}[e^{\rho X}] \leq e^{\frac{\rho^2 \sigma^2}{2}}. \tag{A.1}$$

As noted in Example 5.8 in Vershynin (2012), all bounded or Gaussian random variables are also sub-Gaussian. We can extend the definition of univariate sub-Gaussian distributions to define a general class of multivariate sub-Gaussian distribution on $\mathbb{R}^p$. Let $\Sigma \in \mathbb{R}^{p \times p}$ be a positive semi-definite, real symmetric matrix.

**Definition A.2.** *We say that a random vector $X \in \mathbb{R}^p$ follows a multivariate sub-Gaussian distribution with covariance matrix $\Sigma$ and parameter $\sigma$ if*

1. *$\mathbb{E}[X] = 0$, and $\mathrm{Cov}(X) = \Sigma$.*

2. *For $1 \leq i \leq p$, $\frac{X_i}{\sqrt{\Sigma_{i,i}}}$ is a sub-Gaussian random variable with parameter $\sigma$.*

Our last definition of this section is the sub-exponential distribution (Pollard, 1990).

**Definition A.3.** *A random variable $X$ with mean $0$ is said to follow a sub-exponential distribution with parameters $(\eta, b)$ if,*

$$\mathbb{E}[e^{\lambda X}] \leq e^{\frac{\lambda^2 \eta^2}{2}} \text{ for } |\lambda| < \frac{1}{b}.$$

**Remark A.4.** *It is well established that (see for instance Equation (6) in Honorio & Jaakkola (2014)) the square of a sub-Gaussian random variable with parameter $\sigma$ is sub-exponential with parameters $(4\sqrt{2}\sigma^2, 4\sigma^2)$.*

Next, we recall the matrix Chernoff bound (Theorem 5.1.1 in Tropp (2015)).

**Lemma A.5.** *Let $A_i$ be random matrices such that $\mathbb{E}[A_i] = 0$. If $\|A_i\|_{\infty,\infty} \leq \gamma$ almost surely, then for every $\varepsilon > 0$,*

$$\mathbb{P}\left( \left\| \frac{1}{m} \sum A_i \right\|_{\infty,\infty} \geq \varepsilon \right) \leq 2pe^{-\frac{3\varepsilon^2 m}{8\gamma^2}}.$$

We then recall the following lemma from Lei & Vu (2015).

**Lemma A.6.** *Under the assumptions in Theorem 3.2, let $\tilde{H}$ be the solution to the further constrained problem (3.8). Then $\tilde{H}$ is rank $k$ and unique. Furthermore, there exist $|J| \times k$ orthonormal matrices $U_J$, $\hat{U}_J$ such that:*

1. *$\begin{bmatrix} U_J \\ 0 \end{bmatrix}$ and $\begin{bmatrix} \hat{U}_J \\ 0 \end{bmatrix}$ spans the $k$-dimensional principal subspaces of $\Sigma$ and $S - \rho\tilde{Z}$, respectively.*

2. *There exists a $|J| \times |J|$ orthonormal matrix $Q$ such that*

$$\hat{U}_J = QU_J,$$
$$\|Q - \mathbb{I}_{|J|}\|_F \leq \frac{8\rho|J|}{\lambda_{diff}}.$$

We also recall the following lemma about the principal subspace projection of a matrix. Theorem 1 in Overton & Womersley (1992).

**Lemma A.7.** *Let $A$ be a symmetric matrix with eigenvalues $\gamma_1 \geq \cdots \geq \gamma_p$ and orthonormal eigenvectors $v_1, \ldots, v_p$.*

*$\max_{H \in \mathcal{F}^k} \langle A, H \rangle = \gamma_1 + \cdots + \gamma_k$ and the maximum is achieved by the projector of a $k$-dimensional principal subspace of $A$. Moreover, the maximiser is unique if and only if $\gamma_k > \gamma_{k+1}$.*

We also produce the following lemma along with the proof on the upper bound of the infinite norm of the product of two matrices.

**Lemma A.8.** *Let $A$ and $B$ be two matrices with conformable dimensions. Then,*

$$\|AB\|_{\infty,\infty} \leq \|A\|_{\infty,\infty} \|B\|_{1,\infty}$$

*Proof.*

$$\|AB\|_{\infty,\infty} = \max_{i,j\in[p]} |\sum_{k\in[p]} A_{i,k} B_{k,j}| \leq \max_{i,j,k_0\in[p]} |A_{i,k_0}| |\sum_{k\in[p]} B_{k,j}| \leq \|A\|_{\infty,\infty} \|B\|_{1,\infty}.$$

$\square$

# B   Proofs of Main Theorems

## B.1   Proof of Lemma 3.1

*Proof.* We note that $S^{(i)} - \Sigma^{(i)}$ are i.i.d with $\mathbb{E}\left[S^{(i)} - \Sigma^{(i)}\right] = 0$. We begin by finding an entrywise bound for $\frac{1}{m}\sum\left(S^{(i)} - \Sigma^{(i)}\right)$. Since $S^{(i)} = \frac{1}{p}\sum X_p^{(i)}\left(X_p^{(i)}\right)^T$,

$$\mathbb{P}\left(\left|\frac{1}{mn}\sum_{i,p} X_{k,p}^{(i)} X_{l,p}^{(i)} - \sigma_{k,l}^{(i)}\right| \geq \varepsilon\right).$$

Let $\sigma_k := \max_{1\leq i\leq K} \sigma_{k,k}^{(i)}$, $\tilde{X}_{k,p}^{(i)} := \frac{X_{k,p}^{(i)}}{\sqrt{\sigma_k}}$, $\tilde{\rho}_{ij}^{(k)} := \frac{\sigma_{k,l}^{(i)}}{\sqrt{\sigma_k\sigma_l}}$. We have

$$\mathbb{P}\left(\left|\frac{1}{mn}\sum_{i,p} X_{k,p}^{(i)} X_{l,p}^{(i)} - \sigma_{k,l}^{(i)}\right| \geq \varepsilon\right) = \mathbb{P}\left[4\left|\sum_{i,p}\left(\tilde{X}_{k,p}^{(i)} \tilde{X}_{l,p}^{(i)} - \tilde{\rho}_{k,l}^{(i)}\right)\right| > \frac{4nm\varepsilon}{\sqrt{\sigma_k\sigma_l}}\right]$$

Define $U_{p,k,l}^{(i)} := \tilde{X}_{p,k}^{(i)} + \tilde{X}_{p,l}^{(i)}$, $V_{p,k,l}^{(i)} := \tilde{X}_{p,k}^{(i)} - \tilde{X}_{p,l}^{(i)}$. Then for any $r \in \mathbb{R}$,

$$4\sum_{i,p}\left(\tilde{X}_{k,p}^{(i)} \tilde{X}_{l,p}^{(i)} - \tilde{\rho}_{k,l}^{(i)}\right) = \sum_{k,t}\left\{\left(U_{p,k,l}^{(i)}\right)^2 - 2\left(r + \tilde{\rho}_{k,l}^{(i)}\right)\right\} - \sum_{k,t}\left\{\left(V_{p,k,l}^{(i)}\right)^2 - 2\left(r - \tilde{\rho}_{k,l}^{(i)}\right)\right\} \tag{B.1}$$

Thus,

$$\mathbb{P}\left(\left|\frac{1}{mn}\sum_{i,p} X_{k,p}^{(i)} X_{l,p}^{(i)} - \sigma_{k,l}^{(i)}\right| \geq \varepsilon\right) \leq \mathbb{P}\left[|\sum_{k,t}\left\{\left(U_{t,ij}^{(k)}\right)^2 - 2\left(r + \tilde{\rho}_{ij}^{(k)}\right)\right\}| > \frac{2nm\varepsilon}{\sqrt{\sigma_k\sigma_l}}\right]$$
$$+ \mathbb{P}\left[|\sum_{k,t}\left\{\left(V_{t,ij}^{(k)}\right)^2 - 2\left(r - \tilde{\rho}_{ij}^{(k)}\right)\right\}| > \frac{2nm\varepsilon}{\sqrt{\sigma_k\sigma_l}}\right]. \tag{B.2}$$

Consider the first term

$$\mathbb{P}\left[|\sum_{i,p}\left\{\left(U_{p,k,l}^{(i)}\right)^2 - 2\left(r + \tilde{\rho}_{ij}^{(k)}\right)\right\}| > \frac{2nm\varepsilon}{\sqrt{\sigma_k\sigma_l}}\right],$$

and let $r = \frac{\sigma_{k,k}^{(i)}}{\sigma_k} + \frac{\sigma_{l,l}^{(i)}}{\sigma_l}$. Then,

$$\mathbb{E}\left[\left(U_{p,k,l}^{(i)}\right)^2 - 2\left(r + \tilde{\rho}_{ij}^{(k)}\right)\right] = 0.$$

Let $Z_{p,k,l}^{(i)} := \left(U_{p,k,l}^{(i)}\right)^2 - 2\left(r + \tilde{\rho}_{ij}^{(k)}\right)$. Consequently,

$$\mathbb{P}\left(\left|\sum_{i,p}\left\{\left(U_{p,k,l}^{(i)}\right)^2 - 2\left(r + \tilde{\rho}_{ij}^{(k)}\right)\right\}\right| > \frac{2nm\varepsilon}{\sqrt{\sigma_k\sigma_l}}\right) = \mathbb{P}\left(\sum_{i,p} Z_{p,k,l}^{(i)} > \frac{2nm\varepsilon}{\sqrt{\sigma_k\sigma_l}}\right) \tag{B.3}$$
$$+ \mathbb{P}\left(\sum_{i,p} Z_{p,k,l}^{(i)} < -\frac{2nm\varepsilon}{\sqrt{\sigma_k\sigma_l}}\right)$$

To see that $U_{p,k,l}^{(i)}$ is sub-Gaussian, we note that,

$$\mathbb{E}(e^{\lambda U_{p,k,l}^{(i)}}) = \mathbb{E}\left(e^{\lambda\left(X_{p,k}^{(i)} + X_{p,l}^{(i)}\right)}\right)$$
$$\leq \left[\mathbb{E}e^{2\lambda\tilde{X}_{p,k}^{(i)}}\right]^{0.5}\left[\mathbb{E}e^{2\lambda\tilde{X}_{p,l}^{(i)}}\right]^{0.5}$$
$$\leq e^{\lambda^2\left(\frac{\sigma_{k,k}^{(i)}}{\sigma_k}\sigma\right)^2} e^{\lambda^2\left(\frac{\sigma_{l,l}^{(i)}}{\sigma_l}\sigma\right)^2}.$$

Since $\frac{\sigma_{k,k}^{(i)}}{\sigma_k}$ and $\frac{\sigma_{l,l}^{(i)}}{\sigma_l}$ are upper bounded by 1, we consequently have,

$$\mathbb{E}(e^{\lambda U_{p,k,l}^{(i)}}) \leq e^{2\lambda^2\sigma^2},$$

which shows that $U_{p,k,l}^{(i)}$ is sub-Gaussian with parameter $2\sigma$. We also know, from Section 5.2.4 in Vershynin (2012), that if $U_{p,k,l}^{(i)}$ is a sub-Gaussian random variable with then $\left(U_{p,k,l}^{(i)}\right)^2$ is a sub-exponential random variable. As noted in Section 2 if $X$ is a sub-Gaussian with parameter $\omega$, then, $X^2$ is a sub-exponential distribution with parameters $(4\sqrt{2}\omega^2, 4\omega^2)$. Consequently, if $|t| < \frac{1}{4\omega^2}$,

$$\mathbb{E}[e^{tX^2}] \leq e^{16\omega^4 t^2}.$$

Returning to Equation (B.3), we observe that each $Z_{p,k,l}^{(i)}$ is i.i.d. subexponential with parameter $(16\sqrt{2}\sigma^2, 16\sigma^2)$. Hence for any $\frac{1}{16\sigma^2} > \theta > 0$,

$$\mathbb{P}\left(\sum_{i,p} Z_{p,k,l}^{(i)} > \frac{2nm\varepsilon}{\sqrt{\sigma_k\sigma_l}}\right) = \mathbb{P}\left(\theta\sum_{i,p} Z_{p,k,l}^{(i)} > \theta\frac{2nm\varepsilon}{\sqrt{\sigma_k\sigma_l}}\right)$$
$$= \mathbb{P}\left(e^{\theta\sum_{i,p} Z_{p,k,l}^{(i)}} > e^{\theta\frac{2nm\varepsilon}{\sqrt{\sigma_k\sigma_l}}}\right).$$

Thus, by Markov's inequality, it follows that,

$$\mathbb{P}\left(\sum_{i,p} Z_{p,k,l}^{(i)} > \frac{2nm\varepsilon}{\sqrt{\sigma_k\sigma_l}}\right) \leq \mathbb{E}\left[e^{\theta\sum_{i,p} Z_{p,k,l}^{(i)}}\right] e^{-\theta\frac{2nm\varepsilon}{\sqrt{\sigma_k\sigma_l}}}$$

$$= \prod_{i,p}\mathbb{E}\left[e^{\theta Z_{p,k,l}^{(i)}}\right] e^{-\theta\frac{2nm\varepsilon}{\sqrt{\sigma_k\sigma_l}}}$$

$$\leq \prod_{i,p} e^{512\theta^2\sigma^4} e^{-\theta\frac{2nm\varepsilon}{\sqrt{\sigma_k\sigma_l}}}$$

$$= e^{512nm\theta^2\sigma^4} e^{-\theta\frac{2nm\varepsilon}{\sqrt{\sigma_k\sigma_l}}}$$

$$= e^{512mn\theta^2\sigma^4 - \theta\frac{2nm\varepsilon}{\sqrt{\sigma_k\sigma_l}}}$$

Minimising over $\theta > 0$, we get the derivative of the exponent with respect to $\theta$ to be equal to 0. In other words,

$$512mn\theta\sigma^4 - \frac{2nm\varepsilon}{\sqrt{\sigma_k\sigma_l}} = 0,$$

which in turn implies that whenever

$$\frac{\varepsilon}{512\sigma^4\sqrt{\sigma_k\sigma_l}} < \frac{1}{16\sigma^2},$$

the minimum is achieved when $\theta = \frac{\varepsilon}{512\sigma^4\sqrt{\sigma_k\sigma_l}}$. In other words, if

$$\varepsilon < 32\sigma^2\sqrt{\sigma_k\sigma_l},$$

the minimum is, $e^{-\frac{nm\varepsilon^2}{512\sigma^4\sqrt{\sigma_k\sigma_l}}}$. Consequently, our upper bound can be further refined as,

$$\mathbb{P}\left(\sum_{i,p} Z_{p,k,l}^{(i)} > \frac{2nm\varepsilon}{\sqrt{\sigma_k\sigma_l}}\right) \leq e^{-\frac{nm\varepsilon^2}{512\sqrt{\sigma_k\sigma_l}\sigma^4}}.$$

Since for each $k \in [m]$, $\sigma_k < \max_{i\in[m]}\left\|\Sigma^{(i)}\right\|_{\infty,\infty} < \eta$, this in turn implies that the upper bound is independent of the value of $\Sigma^{(i)}$'s. In particular, this upper bound is distribution free. Hence, for $\varepsilon < 32\eta\sigma^2$,

$$\mathbb{P}\left(\sum_{i,p} Z_{p,k,l}^{(i)} > \frac{2nm\varepsilon}{\sqrt{\sigma_k\sigma_l}}\right) \leq e^{-\frac{nm\varepsilon^2}{512\eta\sigma^4}}.$$

Similarly, the upper bound to $\mathbb{P}\left(\sum_{i,p} Z_{p,k,l}^{(i)} < -\frac{2nm\varepsilon}{\sqrt{\sigma_k\sigma_l}}\right)$, can be achieved; and,

$$\mathbb{P}\left(\sum_{i,p} Z_{p,k,l}^{(i)} < -\frac{2nm\varepsilon}{\sqrt{\sigma_k\sigma_l}}\right) \leq e^{-\frac{nm\varepsilon^2}{512\eta\sigma^4}}.$$

Hence, we have found the entrywise error bound to $\frac{1}{m}\sum\left(S^{(i)} - \Sigma^{(i)}\right)$. To find an upper bound to $\left\|\frac{1}{m}\sum\left(S^{(i)} - \Sigma^{(i)}\right)\right\|_{\infty,\infty}$, we use a union bound.

$$\mathbb{P}\left(\left\|\frac{1}{m}\sum\left(S^{(i)} - \Sigma^{(i)}\right)\right\|_{\infty,\infty} > \varepsilon\right) \leq \sum_{k,l}\mathbb{P}\left(\left|\frac{1}{mn}\sum_{i,p} X_{k,p}^{(i)}X_{l,p}^{(i)} - \sigma_{k,l}^{(i)}\right| \geq \varepsilon\right).$$

$$\leq \frac{p(p+1)}{2} e^{-\frac{nm\varepsilon^2}{512\eta\sigma^4}}.$$

This completes the proof. □

## B.2 Proof of Theorem 3.2

*Proof.* Continue from the setup in Section 3.1. We now have to complete the proof by verifying the remaining 4 steps.

**STEP 1.** We begin by finding a probabilistic upper bound on $\|S - \Sigma\|_{\infty,\infty}$. As seen in Equation (2.4), $S = \frac{1}{m}\sum_{i=1}^{m} S^{(i)}$, where $S^{(i)} = \frac{1}{n}\sum_{j=1}^{n} X_j^{(i)}(X_j^{(i)})^T$ are estimated variance matrix from auxiliary tasks.

$$\|S - \Sigma\|_{\infty,\infty} \le \left\|\frac{1}{m}\sum_{i=1}^{m}\left(S^{(i)} - \Sigma^{(i)}\right)\right\|_{\infty,\infty} + \left\|\frac{1}{m}\sum_{i=1}^{m}\Sigma^{(i)} - \Sigma\right\|_{\infty,\infty} \tag{B.4}$$

$$\text{Term 1} + \text{Term 2}.$$

**TERM 2.** We begin by analysing the second term in Equation (B.4). We note that for any fixed $i$,

$$\left\|\Sigma^{(i)} - \Sigma\right\|_{\infty,\infty} = \left\|R^{(i)}U(\Lambda + D^{(i)})(R^{(i)}U)^T - U\Lambda U^T\right\|_{\infty,\infty}$$

$$\le \left\|R^{(i)}U(\Lambda + D^{(i)})(R^{(i)}U)^T\right\|_{\infty,\infty} + \left\|U\Lambda U^T\right\|_{\infty,\infty}$$

$$= \left\|(R^{(i)} - \mathbb{I}_p + \mathbb{I}_p)U(\Lambda + D^{(i)})U^T(R^{(i)} - \mathbb{I}_p + \mathbb{I}_p)^T\right\|_{\infty,\infty} + \left\|U\Lambda U^T\right\|_{\infty,\infty}$$

$$\le \left\|(R^{(i)} - \mathbb{I}_p)U(\Lambda + D^{(i)})U^T(R^{(i)} - \mathbb{I}_p)^T\right\|_{\infty,\infty} + \left\|U(\Lambda + D^{(i)})U^T(R^{(i)} - \mathbb{I}_p)^T\right\|_{\infty,\infty}$$

$$+ \left\|(R^{(i)} - \mathbb{I}_p)U(\Lambda + D^{(i)})U^T\right\|_{\infty,\infty} + \left\|U\Lambda U^T\right\|_{\infty,\infty}.$$

Applying Lemma A.8 we get,

$$\left\|\Sigma^{(i)} - \Sigma\right\|_{\infty,\infty} \le \|R^{(i)} - \mathbb{I}_p\|_{1,\infty}^2 \left\|U(\Lambda + D^{(i)})U^T\right\|_{\infty,\infty} + 2\|R^{(i)} - \mathbb{I}_p\|_{1,\infty}\left\|U(\Lambda + D^{(i)})U^T\right\|_{\infty,\infty}$$

$$+ \left\|U\Lambda U^T\right\|_{\infty,\infty},$$

which can be upper bounded by,

$$\left(\|R^{(i)} - \mathbb{I}_p\|_{1,\infty} + 1\right)^2 \left(\left\|U\Lambda U^T\right\|_{\infty,\infty} + \left\|UD^{(i)}U^T\right\|_{\infty,\infty}\right). \tag{B.5}$$

Since $U$ is an orthogonal matrix, $\left\|UD^{(i)}U^T\right\|_{\infty,\infty} \le \lambda_1(D^{(i)})$. Then, from Assumption 3, we get,

$$\left\|\Sigma^{(i)} - \Sigma\right\|_{\infty,\infty} \le (\mathbb{C}_R + 1)^2 (\lambda_1(\Sigma) + L)$$

$$= \frac{\lambda^\dagger}{2} \quad \text{almost everywhere.}$$

It is easy to see, by an application of triangle inequality and Jensen's inequality that,

$$\left\|\Sigma^{(i)} - \Sigma - \mathbb{E}[\Sigma^{(i)} - \Sigma]\right\|_{\infty,\infty} \le 2(\mathbb{C}_R + 1)^2 (\lambda_1(\Sigma) + L)$$

$$= \lambda^\dagger$$

We next show that $\Sigma^{(i)}$ are centered random variables. In other words, we show that $\mathbb{E}[\Sigma^{(i)}] = \Sigma$. Observe from Equation (2.1) and Equation (2.3) that

$$\mathbb{E}\left[\Sigma^{(i)}\right] = \mathbb{E}\left[R^{(i)}U(\Lambda + D^{(i)})U^T\left(R^{(i)}\right)^T\right].$$

By linearity of the vec operator, which converts matrices to vectors, by Equation (3.1), the property of the Kronecker product and by Assumption 3, we have

$$\text{vec}\left(\mathbb{E}\left[\Sigma^{(i)}\right]\right) = \text{vec}\left(\mathbb{E}\left[R^{(i)}U(\Lambda + D^{(i)})U^T\left(R^{(i)}\right)^T\right]\right)$$

$$= \mathbb{E}\left[\text{vec}\left(R^{(i)}U(\Lambda + D^{(i)})U^T\left(R^{(i)}\right)^T\right)\right]$$

$$= \mathbb{E}\left[\left(R^{(i)} \otimes R^{(i)}\right)\text{vec}\left(U(\Lambda + D^{(i)})U^T\right)\right]$$

$$= \mathbb{E}\left[R^{(i)} \otimes R^{(i)}\right]\text{vec}\left(\mathbb{E}\left[U(\Lambda + D^{(i)})U^T\right]\right)$$

$$= \mathbb{E}\left[R^{(i)} \otimes R^{(i)}\right]\text{vec}\left(\Sigma + U\mathbb{E}[D^{(i)}]U^T\right)$$

$$= \mathbb{E}\left[R^{(i)} \otimes R^{(i)}\right]\text{vec}(\Sigma)$$

$$= \text{vec}(\Sigma).$$

Thus, we get

$$\mathbb{E}[\Sigma^{(i)}] = \Sigma. \tag{B.6}$$

Now, putting $A_i = \Sigma^{(i)} - \Sigma - \mathbb{E}[\Sigma^{(i)} - \Sigma]$ applying Lemma A.5, we get that for any $\varepsilon > 0$

$$\mathbb{P}\left(\left\|\frac{1}{m}\sum_{i=1}^{m}\left(\Sigma^{(i)} - \Sigma - \mathbb{E}[\Sigma^{(i)} - \Sigma]\right)\right\|_{\infty,\infty} > \varepsilon\right) \le 2pe^{-\frac{3\varepsilon^2 m}{8(\lambda^\dagger)^2}}.$$

Following Equation (B.6), we set $\mathbb{E}[\Sigma^{(i)} - \Sigma] = 0$ and let

$$\varepsilon = 4\lambda^\dagger\sqrt{\frac{\log(p)}{m}}.$$

Now by an application of triangle inequality, we get

$$\frac{1}{m}\sum_{i=1}^{m}\left\|\Sigma^{(i)} - \Sigma\right\|_{\infty,\infty} \ge \left\|\frac{1}{m}\sum_{i=1}^{m}\left(\Sigma^{(i)} - \Sigma\right)\right\|_{\infty,\infty} \ge 4\lambda^\dagger\sqrt{\frac{\log(p)}{m}}$$

with probability at most $\frac{2}{p^2}$.

TERM 1. In this part of the proof, we provide an upper bound for the first term from Equation (B.4). We begin by finding an upper bound to $\left\|\Sigma^{(i)}\right\|_{\infty,\infty}$. Similarly, as to the derivation of Equation (B.5) we can write,

$$\left\|\Sigma^{(i)}\right\|_{\infty,\infty} = \left\|R^{(i)}U(\Lambda + D^{(i)})(R^{(i)}U)^T\right\|_{\infty,\infty}$$

$$= \left\|(R^{(i)} - \mathbb{I}_p + \mathbb{I}_p)U(\Lambda + D^{(i)})U^T(R^{(i)} - \mathbb{I}_p + \mathbb{I}_p)^T\right\|_{\infty,\infty}$$

$$\le \left\|(R^{(i)} - \mathbb{I}_p)U(\Lambda + D^{(i)})U^T(R^{(i)} - \mathbb{I}_p)^T\right\|_{\infty,\infty} + \left\|U(\Lambda + D^{(i)})U^T(R^{(i)} - \mathbb{I}_p)^T\right\|_{\infty,\infty}$$

$$+ \left\|(R^{(i)} - \mathbb{I}_p)U(\Lambda + D^{(i)})U^T\right\|_{\infty,\infty}.$$

Applying Lemma A.8 we can decompose the second and the third terms. This yields us,

$$\left\|\Sigma^{(i)}\right\|_{\infty,\infty} \le \|R^{(i)} - \mathbb{I}_p\|_{1,\infty}^2\left\|U(\Lambda + D^{(i)})U^T\right\|_{\infty,\infty} + 2\|R^{(i)} - \mathbb{I}_p\|_{1,\infty}\left\|U(\Lambda + D^{(i)})U^T\right\|_{\infty,\infty}$$

$$\le \|R^{(i)} - \mathbb{I}_p\|_{1,\infty}^2\left\|U(\Lambda + D^{(i)})U^T\right\|_{\infty,\infty} + 2\|R^{(i)} - \mathbb{I}_p\|_{1,\infty}\left\|U(\Lambda + D^{(i)})U^T\right\|_{\infty,\infty}$$

$$+ \left\|U\Lambda U^T\right\|_{\infty,\infty}$$

$$= \lambda^\dagger,$$

with the last inequality following trivially by the fact that $\left\|U\Lambda U^T\right\|_{\infty,\infty} \geq 0$. By using Lemma 3.1 we get,

$$\mathbb{P}\left(\left\|\frac{1}{m}\sum\left(S^{(i)} - \Sigma^{(i)}\right)\right\|_{\infty,\infty} \geq \varepsilon\right) \leq \frac{p(p+1)}{2}e^{-\frac{nm\varepsilon^2}{512\sigma^4\lambda^\dagger}}.$$

putting $\varepsilon = 16\sqrt{2\sigma^4\lambda^\dagger\frac{\log(p+1)}{nm}}$, we get with probability $1 - \frac{1}{2(p+1)^2}$,

$$\mathbb{P}\left(\left\|\frac{1}{m}\sum\left(S^{(i)} - \Sigma^{(i)}\right)\right\|_{\infty,\infty} \leq 16\sqrt{2\sigma^4\lambda^\dagger\frac{\log(p+1)}{nm}}\right).$$

Thus,

$$\mathbb{P}\left(\|S - \Sigma\|_{\infty,\infty} > \max\left\{4\lambda^\dagger\sqrt{\frac{\log(p)}{m}}, 16\sqrt{2\sigma^4\lambda^\dagger\frac{\log(p+1)}{nm}}\right\}\right) \leq \frac{2}{p^2} + \frac{1}{2(p+1)^2}.$$

Since,

$$\frac{8|J|}{\lambda_k - \lambda_{k+1}}\|\Sigma_{J^c,J}\|_{2,\infty} < 1,$$

there exists a constant $\alpha \in (0,1)$ such that if,

$$\rho > \alpha\max\left\{4\lambda^\dagger\sqrt{\frac{\log(p)}{m}}, 16\sqrt{2\sigma^4\lambda^\dagger\frac{\log(p+1)}{nm}}\right\},$$

then with probability at least $1 - \frac{2}{p^2} - \frac{1}{2(p+1)^2}$,

$$\rho^{-1}\|S - \Sigma\|_{\infty,\infty} + \frac{8|J|}{\lambda_{diff}}\|\Sigma_{J^c,J}\|_{2,\infty} \leq 1. \tag{B.7}$$

It only remains to choose an appropriate matrix $Q$. By the application of Lemma A.6 there exists a $Q$ matrix such that

$$\|Q - \mathbb{I}_{|J|}\|_F \leq \frac{8\rho|J|}{\lambda_{diff}},$$

and

$$U_J = Q\hat{U}_J,$$

where $U_J$ and $\hat{U}_J$ are defined as in Lemma A.6. We select this $Q$ for the remainder of the proof.

**STEP 2.** In this section of the proof, we show that our proposed estimator is feasible, unique, and optimal, i.e., satisfies the KKT conditions (3.5,3.6,3.7).

FEASIBLITY. The feasibility of $\hat{H}$ is obvious from construction. To check the feasibility of $\hat{Z}$, we need to verify that for a given choice of $\rho$,

$$\hat{Z}_{ij} \in [-1,1], \ \forall (i,j) \in (J \times J)^c.$$

Therefore, it is sufficient to verify

$$\frac{1}{\rho}\left[|S_{ij} - \Sigma_{ij}| + |\Sigma_{ij} - \langle Q_{i*}, \Sigma_{J,j}\rangle|\right] \leq 1.$$

By an application of Cauchy-Schwarz inequality, we get

$$\frac{1}{\rho}\left[|S_{ij} - \Sigma_{ij}| + |\Sigma_{ij} - \langle Q_{i*}, \Sigma_{J,j}\rangle|\right] \leq \frac{1}{\rho}\left[|\ \|S - \Sigma\|_{\infty,\infty} + \|Q - \mathbb{I}_{|J|}\|_F\|\Sigma\|_{2,\infty}\right]$$

$$\leq \frac{1}{\rho}\left[\|S - \Sigma\|_{\infty,\infty} + \frac{8\rho|J|}{\lambda_{diff}}\|\Sigma\|_{2,\infty}\right].$$

Thus, with probability at least $1 - \frac{2}{p^2} - \frac{1}{2(p+1)^2}$,

$$\rho^{-1}\|S - \Sigma\|_{\infty,\infty} + \frac{8|J|}{\lambda_{diff}}\|\Sigma_{J^c,J}\|_{2,\infty} \leq 1. \tag{B.8}$$

OPTIMALITY. $\hat{H}$ only has non-zero entries in the sub-matrix $\hat{H}_{J,J}$. By construction $(\hat{H}_{i,j}, \hat{Z}_{i,j}) = (\tilde{H}_{i,j}, \tilde{Z}_{i,j})$, $\forall(i,j) \in J \times J$. From the optimality of $(\tilde{H}, \tilde{Z})$, Equation (3.5) is satisfied.

The fact that Equation (3.6) is satisfied follows from the optimality of $(\tilde{H}, \tilde{Z})$ when $(i,j) \in J \times J$ and feasibility of $\tilde{Z}$ otherwise.

Note that $\hat{U}_J$ spans the $k$-dimensional principal subspace of $S - \rho\tilde{Z}$. We now show that $\hat{U}$ spans one (not necessarily principal) $k$-dimensional subspace of $S - \rho\hat{Z}$. In other words, it is enough to show that, for some diagonal matrix $D_0$,

$$(S - \rho\hat{Z})\begin{bmatrix} \hat{U}_J \\ 0 \end{bmatrix} = D\begin{bmatrix} \hat{U}_J \\ 0 \end{bmatrix}. \tag{B.9}$$

Analysing the term on the left-hand side of the previous equation, we get,

$$(S - \rho\hat{Z})\begin{bmatrix} \hat{U}_J \\ 0 \end{bmatrix} = \begin{bmatrix} S_{J,J} - \rho\tilde{Z}_{J,J} & Q\Sigma_{J,J^c} \\ \Sigma_{J^c,J}Q^T & \Sigma_{J^c,J^c} + \mathbb{D}(S_{J^c,J^c} - \Sigma_{J^c,J^c}) \end{bmatrix}\begin{bmatrix} \hat{U}_J \\ 0 \end{bmatrix}$$
$$= \begin{bmatrix} (S_{J,J} - \rho\tilde{Z}_{J,J})\hat{U}_J \\ \Sigma_{J^c,J}Q^T\hat{U}_J \end{bmatrix}$$
$$= \begin{bmatrix} (S_{J,J} - \rho\tilde{Z}_{J,J})\hat{U}_J \\ \Sigma_{J^c,J}Q^TQU_J \end{bmatrix}$$

Recall from Lemma A.6 that there exists a diagonal matrix $D_1$ such that $(S_{J,J} - \rho\tilde{Z}_{J,J})\hat{U}_J = D_1\hat{U}_J$. Also, the fact that $\begin{bmatrix} U_J \\ 0 \end{bmatrix}$ spans the $k$-dimensional principal subspace of $\Sigma$ implies that $\Sigma_{J^c,J}U_J = 0$. Therefore,

$$\begin{bmatrix} (S_{J,J} - \rho\tilde{Z}_{J,J})\hat{U}_J \\ \Sigma_{J^c,J}Q^TQU_J \end{bmatrix} = \begin{bmatrix} D_1\hat{U}_J \\ \Sigma_{J^c,J}U_J \end{bmatrix}$$
$$= \begin{bmatrix} D_1\hat{U}_J \\ 0 \end{bmatrix}$$
$$= \begin{bmatrix} D_1\hat{U}_J \\ 0 \end{bmatrix}$$
$$= \begin{bmatrix} D_1 & 0 \\ 0 & 0 \end{bmatrix}\begin{bmatrix} \hat{U}_J \\ 0 \end{bmatrix}$$

Setting

$$D_0 = \begin{bmatrix} D_1 & 0 \\ 0 & 0 \end{bmatrix}$$

we successfully show that Equation (B.9) holds. We will further prove that $diag(D_1)$ are the first $k$ eigenvalues of $(S - \rho\hat{Z})$, and $\lambda_k(S - \rho\hat{Z}) - \lambda_{k+1}(S - \rho\hat{Z}) > 0$. Then uniqueness will follow by Lemma A.6.

$$(S - \rho\hat{Z}) = \begin{bmatrix} S_{JJ} - \rho\tilde{Z}_{JJ} - Q\Sigma_{JJ}Q^T & 0 \\ 0 & \Sigma_{J^c,J^c} + \mathbb{D}(S_{J^c,J^c} - \Sigma_{J^c,J^c}) \end{bmatrix} + \begin{bmatrix} Q\Sigma_{JJ}Q^T & Q\Sigma_{JJ^c} \\ \Sigma_{J^c,J}Q^T & \Sigma_{J^c,J^c} \end{bmatrix}$$
$$= \begin{bmatrix} S_{JJ} - \rho\tilde{Z}_{JJ} - Q\Sigma_{JJ}Q^T & 0 \\ 0 & \Sigma_{J^c,J^c} + \mathbb{D}(S_{J^c,J^c} - \Sigma_{J^c,J^c}) \end{bmatrix} + \begin{bmatrix} Q & 0 \\ 0 & I \end{bmatrix} \times \Sigma \times \begin{bmatrix} Q^T & 0 \\ 0 & I \end{bmatrix}$$

However, since we just proved that $(S - \rho\widehat{Z})$ satisfies the conditions of Proposition 4.1, the problem reduces to finding an upper bound to $\lambda_l(S_{J,J} - \rho\tilde{Z}_{J,J} - Q\Sigma_{J,J}Q^T)$ for each $l \in [|J|]$. Note that,

$$
\begin{aligned}
\lambda_l(S_{J,J} - \rho\tilde{Z}_{J,J} - Q\Sigma_{J,J}Q^T) &\leq \lambda_l(S_{J,J} - \rho\tilde{Z}_{J,J} - \Sigma_{J,J}) + \lambda_l(\Sigma_{J,J} - Q\Sigma_{J,J}Q^T) \\
&\leq \|S_{J,J} - \rho\tilde{Z}_{J,J} - \Sigma_{J,J}\|_F + \|\Sigma_{J,J} - Q\Sigma_{J,J}Q^T\|_F \\
&\leq \|S_{J,J} - \rho\tilde{Z}_{J,J} - \Sigma_{J,J}\|_F + 2\|\Sigma_{J,J}\|_F \times \|Q - I\|_F \\
&\leq 2\rho|J| + 2\lambda_1(\Sigma) \times 8\rho|J|/(\lambda_k - \lambda_{k+1}).
\end{aligned}
$$

Where the last inequality follows from the combination of part 1 and Lemma A.6. The rest of the proof follows since for a large enough $\rho$ we have,

$$
4\rho|J| + \frac{32\lambda_1(\Sigma)\rho|J|}{(\lambda_k - \lambda_{k+1})} \leq \lambda_k - \lambda_{k+1}.
$$

This completes the proof of step 2.

**STEP 3.** In this section of the proof, we prove that $\left\|\hat{H} - \Pi\right\|_{\infty,\infty} < 2\frac{\rho|J|}{\lambda_{diff}}$. By step 2, we have,

$$
\begin{aligned}
\left\|\hat{H} - \Pi\right\|_{\infty,\infty} &= \left\|\begin{bmatrix} \hat{U}_J\hat{U}_J^T & 0 \\ 0 & 0 \end{bmatrix} - \begin{bmatrix} U_JU_J^T & 0 \\ 0 & 0 \end{bmatrix}\right\|_{\infty,\infty} \\
&= \left\|\begin{bmatrix} QU_JU_J^TQ^T & 0 \\ 0 & 0 \end{bmatrix} - \begin{bmatrix} U_JU_J^T & 0 \\ 0 & 0 \end{bmatrix}\right\|_{\infty,\infty} \\
&= \|QU_JU_J^TQ^T - U_JU_J^T\|_{\infty,\infty} \\
&\leq \lambda_1(U_JU_J^T)\|\mathbb{I} - Q\|_F \\
&\leq \frac{8|J|\rho}{\lambda_{diff}}.
\end{aligned}
$$

Setting $\mathbb{C}_\pi = \frac{8|J|}{\lambda_{diff}}$, completes the proof of the upper bound for $\left\|\hat{H} - \Pi\right\|_{\infty,\infty}$.

**STEP 4.** Set

$$
\rho < \min_{i \in J} \frac{\Pi_{i,i}\lambda_{diff}}{16|J|}.
$$

The rest of the proof now follows from the step 3. $\qquad\square$

### B.3 Proof of Theorem 3.3

We start with a technical lemma.

**Lemma B.1.** $\left\|R^{(i)}(J) - \mathbb{I}_p\right\|_{1,\infty} \leq 2\sqrt{p}$, $|R^{(i)}(J)| \leq 1/\sqrt{p}$ and $\left(R^{(i)}(J)\right)^2$ is an upper triangular matrix such that,

$$
\left(R^{(i)}(J')\right)^2 = \begin{bmatrix} 1 & 2p^{-1/2} - 2p^{-1} & \dots & \dots & 2p^{-1/2} - 2p^{-1} \\ 0 & 1 & \dots & \dots & 0 \\ \vdots & \vdots & \vdots & \vdots & \vdots \\ 0 & 0 & \dots & \dots & 1 \end{bmatrix}.
$$

*Proof.* First, we note that,

$$
\begin{aligned}
\left\|R^{(i)}(J') - \mathbb{I}_p\right\|_{1,\infty} &= \sum_{l=2}^{p} |p^{-1/2} - p^{-1}| \\
&= \frac{p}{\sqrt{p}} - 1 \\
&\leq 2\sqrt{p}.
\end{aligned}
$$

This proves the first part. To observe the second part, note that $R^{(i)}(J')$ is an upper triangular matrix such that,

$$\left(R^{(i)}(J')\right) = \begin{bmatrix} 1 & p^{-1/2}-p^{-1} & \cdots & \cdots & p^{-1/2}-p^{-1} \\ 0 & 1 & \cdots & \cdots & 0 \\ \vdots & \vdots & \vdots & \vdots & \vdots \\ 0 & 0 & \cdots & \cdots & 1 \end{bmatrix}.$$

Since $p$ is even, the determinant can be calculated as

$$\left|R^{(i)}(J')\right| = 1 + \sum_{l=2}^{p}(-1)^l\left(\frac{1}{\sqrt{p}}-\frac{1}{p}\right) = \frac{1}{\sqrt{p}}-\frac{1}{p} \le \frac{1}{\sqrt{p}}$$

By simple matrix multiplication, we can verify that

$$\left(R^{(i)}(J')R^{(i)}(J')\right) = \begin{bmatrix} 1 & 2\times(-1)p^{-1/2}-p^{-1} & \cdots & \cdots & 2\times(-1)^l p^{-1/2}-p^{-1} \\ 0 & 1 & \cdots & \cdots & 0 \\ \vdots & \vdots & \vdots & \vdots & \vdots \\ 0 & 0 & \cdots & \cdots & 1 \end{bmatrix}. \tag{B.10}$$

To invert $R^{(i)}(J')R^{(i)}(J')$, observe that

$$\begin{bmatrix} 1 & 2\times(-1)^2 p^{-1/2}-p^{-1} & \cdots & \cdots & 2\times(-1)^{p+1}p^{-1/2}-p^{-1} \\ 0 & 1 & \cdots & \cdots & 0 \\ \vdots & \vdots & \vdots & \vdots & \vdots \\ 0 & 0 & \cdots & \cdots & 1 \end{bmatrix} R^{(i)}(J')R^{(i)}(J')$$

$$= \begin{bmatrix} 1 & 2\times(-1)p^{-1/2}-p^{-1}+2(-1)^2 p^{-1/2}-p^{-1} & \cdots & \cdots & 2\times(-1)^p p^{-1/2}-p^{-1}+2(-1)^{p+1}p^{-1/2}-p^{-1} \\ 0 & 1 & \cdots & \cdots & 0 \\ \vdots & \vdots & \vdots & \vdots & \vdots \\ 0 & 0 & \cdots & \cdots & 1 \end{bmatrix}$$

$$= \mathbb{I}_p$$

which proves our lemma. $\square$

Next, we formally prove Theorem 3.3.

*Proof.* To show this part, let $p$ be an odd number, let $g_p$ be $2p$. Set $U = \mathbb{I}_p$, and let the true support be $J_*$. It is easy to see that $|J_*| = k$. Observe that there are $\binom{p}{k}$ many choices for $J_*$ and assume that nature chooses one uniformly. Fix one such choice $J$. For that $J$, define the diagonal matrix $\Lambda(J)$ as follows:

$$\Lambda_{jj}(J) = \begin{cases} \frac{k+1}{k}; & j \in J \\ 1; & j \notin J. \end{cases}$$

Next, recall that there are $\binom{p}{k}$ many choices for $J$. For first half of those (denoted by $J^{(1)}$), assign $diag(R^{(i)}(J)) = (1,\ldots,1))$ and for $l$ in $2,\ldots,p$ set $R_{1,l}^{(i)}(J) = p^{-1/2}-p^{-1}$. Every other cell is set to be 0. For the other half (denoted by $J^{(2)}$), set them to be $\mathbb{I}_p$. We can see that $\left\|R^{(i)}\right\|_{1,\infty} \le g_p$. Next, define $D^{(i)}(J)$ as follows: Let $s$ be the smallest element of $i \ modulo \ k+1$, $i \in J$ and $t$ be the smallest element of $i \ modulo \ p-k+1$, $i \in J^c$. Then,

$$D_{jj}^{(i)}(J) = \begin{cases} -\frac{1}{k}; & j = s \\ \frac{1}{k}; & j = t \\ 0; & otherwise \end{cases}$$

Note that, $\Lambda(J)$ and $\Lambda(J) + D^{(i)}(J)$ has $k$ values $(k+1)/k$ and $p-k$ values $1$. Observe that,

$$\Sigma^{(i)}(J) = \Lambda(J) + D^{(i)}(J) \tag{B.11}$$

It follows that, $|\Sigma^{(i)}(J)| = \left(\frac{k+1}{k}\right)^k$. Let $X_j^{(i)}(J) \sim N(0, \Sigma^{(i)}(J))$. Observe that the KL divergence between two multivariate Gaussian random variables of dimension $p$ can be written as

$$KL(N(0,\Sigma_1)||N(0,\Sigma_2)) = \frac{1}{2}\left(\log\frac{|\Sigma_1|}{|\Sigma_2|} + \operatorname{tr}(\Sigma_2^{-1}\Sigma_1) - p\right)$$

Substituting, $\Sigma^{(i)}(J)$ in place of $\Sigma_1$ and $\Sigma^{(i)}(J')$ in place of $\Sigma_2$, and assume that nature chose $J \in J^{(1)}$ and $J' \in J^{(2)}$. It follows that

$$KL\left(N(0,\Sigma^{(i)}(J))||N(0,\Sigma^{(i)}(J'))\right) = \frac{1}{2}\left(\log\frac{\Lambda(J') + D^{(i)}(J')|}{|R^{(i)}(J)|^2||\Lambda(J) + D^{(i)}(J)|}\right.$$
$$\left. + \operatorname{tr}\left(\left(\Lambda(J') + D^{(i)}(J')\right)^{-1}\left(R^{(i)}(J)\left(\Lambda(J) + D^{(i)}\right)(J)R^{(i)}(J)\right)\right) - p\right)$$

Observe that $|\Lambda(J') + D^{(i)}(J')| = |\Lambda(J) + D^{(i)}(J)| = \left(\frac{k+1}{k}\right)^k$. Thus,

$$\log\frac{|R^{(i)}(J')|^2|\Lambda(J') + D^{(i)}(J')|}{|\Lambda(J) + D^{(i)}(J)|} = \log|R^{(i)}(J)^2| = \log\left|R^{(i)}(J')\right|^2$$

Furthermore, observe that $\left(\Lambda(J') + D^{(i)}(J')\right)$ is a diagonal matrix with entries at least $1$. Therefore,

$$\operatorname{tr}\left(\left(\Lambda(J') + D^{(i)}(J')\right)^{-1}\left(R^{(i)}(J)\left(\Lambda(J) + D^{(i)}(J)\right)R^{(i)}(J)\right)\right) \le \operatorname{tr}\left(\left(R^{(i)}(J)R^{(i)}(J)\right)\left(\Lambda(J) + D^{(i)}(J)\right)\right)$$

Lemma B.1 implies that $\left(R^{(i)}(J)\right)^2$ is an upper triangular matrix with diagonal entries $1$. Since $\left(\Lambda(J) + D^{(i)}(J)\right)$ is a diagonal matrix, it now follows that,

$$\operatorname{tr}\left(\left(R^{(i)}(J)\right)^2\left(\Lambda(J) + D^{(i)}(J)\right)\right) = \operatorname{tr}\left(\Lambda(J) + D^{(i)}(J)\right).$$

We now evaluate the right hand side of the previous expression to get

$$\operatorname{tr}\left(\Lambda(J) + D^{(i)}(J)\right) = k\left(\frac{k+1}{k}\right) + p - k.$$

Therefore,

$$KL\left(N(0,\Sigma^{(i)}(J))||N(0,\Sigma^{(i)}(J'))\right) \le \log\left|R^{(i)}(J)\right|^{-2} + k\left(\frac{k+1}{k}\right) + p - k - p = 1$$

where the inequality follows from the fact that $k$ many of $\Lambda(J) + D^{(i)}(J)$'s entries were $(k+1)/k$ and the rest were $1$. It follows from the explicit expression of $\left[R^{(i)}(J')\right]^2$ in Equation (B.10) that $\left|\left[R^{(i)}(J')\right]\right|^2 = (p^{-1})$. We now have that,

$$KL\left(N(0,\Sigma^{(i)}(J))||N(0,\Sigma^{(i)}(J'))\right) \le \log\left(p^{-1}\right) + 1 \le p^{-1},$$

where the final inequality follows from the fact that $\log(x) \le x - 1$ for all positive $x$ Using Fano's inequality combined with data processing inequality (Fano, 1949) that,

$$\mathbb{P}\left(\hat{J} \ne J_*\right) \ge 1 - \frac{1}{2}\frac{1}{\binom{p}{k}^2}\sum_{J,J'}\sum_{m,n}\frac{KL\left(N(0,\Sigma^{(i)}(J))||N(0,\Sigma^{(i)}(J'))\right)}{\log\binom{p}{k}} - \frac{1}{2}\frac{\log 2}{\log\binom{p}{k}}$$

$$\ge 1 - \frac{1}{2}\frac{1}{\binom{p}{k}^2}\sum_{\substack{J\in J^{(1)}\\J'\in J^{(2)}}}\sum_{m,n}\frac{KL\left(N(0,\Sigma^{(i)}(J))||N(0,\Sigma^{(i)}(J'))\right)}{\log\binom{p}{k}} - \frac{1}{2}\frac{\log 2}{\log\binom{p}{k}}$$

$$\ge 1 - \frac{1}{4}\frac{mn}{p\log\binom{p}{k}} - \frac{1}{2}\frac{\log 2}{\log\binom{p}{k}}$$

$$\ge 1 - \frac{1}{4}\frac{mn}{pk\log p - k\log k} - \frac{1}{2}\frac{\log 2}{k\log p - k\log k},$$

where the last inequality follows since $(p/k)^k \leq \binom{p}{k}$. Let $p$ be large enough such that

$$\frac{1}{2}\frac{\log 2}{k \log p - k \log k} \leq \frac{1}{2}.$$

Then,

$$\mathbb{P}\left(\hat{J} \neq J_*\right) \geq \frac{1}{4}$$

whenever,

$$n \leq \frac{pk \log p - k \log k}{m}.$$

Recall that $g_p$ was set to be $2p$. This implies that,

$$\mathbb{P}\left(\hat{J} \neq J_*\right) \geq \frac{1}{4}$$

whenever,

$$n \leq \frac{g_p k \log p - k \log k}{2m}.$$

This completes the proof. □

### B.4 Proof of Proposition 4.1

*Proof.* Let $\Pi^{(m+1)}$ be the $k$-component PC matrix corresponding to $\Sigma^{(m+1)}$. Hence,

$$\Pi^{(m+1)} = \sum_{i=1}^{k} U_{*,i}^{(m+1)}\left(U_{*,i}^{(m+1)}\right)^T.$$

We also know from the recovered support union that $\Pi_{i,i}^{(m+1)} = 0$ whenever $i \notin J$. However,

$$\Pi_{i,i}^{(m+1)} = \sum_{j=1}^{k}\left(U_{j,i}^{(m+1)}\right)^2.$$

Hence, $\forall j \in J^c, u_{i,j}^{(m+1)} = 0$. Therefore,

$$U_i^{(m+1)} = \begin{pmatrix} U_{J,i}^{(m+1)} \\ 0 \end{pmatrix}.$$

Moreover, we also know that

$$\Sigma^{(m+1)} U_i^{(m+1)} = \lambda_i U_i^{(m+1)}.$$

In other words,

$$\begin{bmatrix} \Sigma_{J,J}^{(m+1)} & \Sigma_{J,J^c}^{(m+1)} \\ \Sigma_{J^c,J}^{(m+1)} & \Sigma_{J^c,J^c}^{(m+1)} \end{bmatrix}\begin{pmatrix} U_{J,i}^{(m+1)} \\ 0 \end{pmatrix} = \lambda_i \begin{pmatrix} U_{J,i}^{(m+1)} \\ 0 \end{pmatrix}.$$

Which in turn implies,

$$\Sigma_{J,J}^{(m+1)} U_{J,i}^{(m+1)} = \lambda_i U_{J,i}^{(m+1)}.$$

This completes the proof. □

### B.5 Proof of Theorem 4.2

*Proof.* As given, $\Sigma^{(m+1)}$ is the novel covariance matrix with the following block structure,

$$\Sigma^{(m+1)} = \begin{bmatrix} \Sigma_{J,J}^{(m+1)} & \Sigma_{J,J^c}^{(m+1)} \\ \Sigma_{J^c,J}^{(m+1)} & \Sigma_{J^c,J^c}^{(m+1)} \end{bmatrix}.$$

For $1 \le i \le k$, let $U_i^{(m+1)}$ be the eigenvectors corresponding to $\Sigma^{(m+1)}$, and $\lambda_i^{(m+1)}$ be the corresponding eigenvalues. Let $J$ be the recovered support union. $U_{J,i}^{(m+1)}$ denotes those coordinates of $U_i^{(m+1)}$ that belong to set $J$. Similarly to the proof of Theorem 3.2, if the penalty parameter $\rho$ satisfies,

$$\rho^{-1}\|S_{J,J}^{(m+1)} - \Sigma_{J,J}^{(m+1)}\|_{\infty,\infty} + \frac{8|J|^{(m+1)}}{\lambda_k^{(m+1)} - \lambda_{k+1}^{(m+1)}}\|\Sigma_{(J^{(m+1)})^c J^{(m+1)}}\|_{2,\infty} \le 1 \quad \text{and}$$

$$4\rho|J^{(m+1)}|\left(1 + \frac{8\lambda_1^{(m+1)}}{\lambda_k^{(m+1)} - \lambda_{k+1}^{(m+1)}}\right) < \lambda_k^{(m+1)} - \lambda_{k+1}^{(m+1)},$$

then the solution $\widehat{H}^{(m+1)}$ of Equation (4.2) is unique and satisfies $supp(\widehat{H}^{(m+1)}) \subseteq J^{(m+1)}$. It is noteworthy that if for each $j \in [n^{(m+1)}]$, $X_j^{(m+1)}$ is a $p$-dimensional multivariate sub-Gaussian distribution with covariance matrix $\Sigma$ and parameter $\sigma$, then for any set of sub-indices $J$, $X_{j,J}^{(m+1)}$ is a $|J|$-dimensional multivariate sub-Gaussian distribution with covariance matrix $\Sigma_{J,J}$ and parameter $\sigma$. Following from Equation (4.1), we can also see that by construction

$$S_{J,J}^{(m+1)} = \frac{1}{n^{(m+1)}} \sum_{l=1}^{n^{(m+1)}} X_{l,J}^{(m+1)}\left(X_{l,J}^{(m+1)}\right)^T.$$

By using Lemma 3.1, we know that as long as $\varepsilon \le 32\lambda_1\left(\Sigma^{(m+1)}\right)\sigma^2$,

$$\mathbb{P}\left(\left\|\left(S^{(m+1)} - \Sigma^{(m+1)}\right)\right\|_{\infty,\infty} \ge \varepsilon\right) \le \frac{|J|(|J|+1)}{2}e^{-\frac{n\varepsilon^2}{512\sigma^4\lambda_1\left(\Sigma^{(m+1)}\right)}}.$$

As before, let $\alpha^{(m+1)} = 1 - \frac{8|J|^{(m+1)}}{\lambda_k^{(m+1)} - \lambda_{k+1}^{(m+1)}}\|\Sigma_{(J^{(m+1)})^c J^{(m+1)}}\|_{2,\infty}$. By Assumption 5, $\alpha^{(m+1)} \in (0,1)$. Therefore, as long as,

$$n^{\frac{1}{3}} > \max\left\{\frac{1}{32\lambda_1\left(\Sigma^{(m+1)}\sigma^2\right)}, \left(\frac{\lambda_k^{(m+1)} - \lambda_{k+1}^{(m+1)}}{4|J^{(m+1)}|\left(1 + \frac{8\lambda_1^{(m+1)}}{\lambda_k^{(m+1)} - \lambda_{k+1}^{(m+1)}}\right)}\right)^{-1}\right\} \quad \text{and,}$$

$$\frac{1}{n^{1/3}\alpha^{(m+1)}} < \rho < \frac{\lambda_k^{(m+1)} - \lambda_{k+1}^{(m+1)}}{4|J^{(m+1)}|\left(1 + \frac{8\lambda_1^{(m+1)}}{\lambda_k^{(m+1)} - \lambda_{k+1}^{(m+1)}}\right)},$$

with probability $1 - \frac{|J|(|J|+1)}{2}e^{-\frac{n^{\frac{1}{3}}}{512\sigma^4\lambda_1\left(\Sigma^{(m+1)}\right)}}$, we have $supp(\Pi^{(m+1)}) = J^{(m+1)}$. $\qquad\square$

## C Additional Experiments

### C.1 Synthetic Experiments with Other Distributions

To validate our results, we first generate random matrices $\Sigma^{(i)}$ for each $i \in [m]$ and then generate $X_j^{(i)}$ for each $j \in [n]$. We now describe the generative procedure for $\Sigma^{(i)}$. We first produce a random orthogonal matrix $U$

where $U_{l,*}$ is supported on $[J]$ for each $l \in [k]$. We then produce a random diagonal matrix with each entry produced uniformly from $[0, 1]$ and add $500$ to the first $k$ many diagonal entries to produce $\Lambda$. $\Sigma = U\Lambda(U)^T$. For each $i \in [m]$, we generate randomised uniform white noise matrices $\mathcal{A}^{(i)}$ $\mathcal{A}_{l,j}^{(i)} \overset{iid}{\sim} Uniform(0,1)$. $R^{(i)}$ is produced by adding scaled $\mathcal{A}^{(i)}$'s to $\mathbb{I}_p$; i.e., $R^{(i)} = \mathcal{A}^{(i)}/p^2 + \mathbb{I}_p$. We generate $D^{(i)}$'s, as white noise matrices where for each $l, j \in [p]; l > j, D_{l,j}^{(i)}$ is generated uniformly from $(0, 1)$. This ensures Assumption 3 is satisfied. $\Sigma^i$ is then generated as, $\Sigma^{(i)} = R^{(i)}U(\Lambda + D^{(i)})U^T \left(R^{(i)}\right)^T$. We compute the empirical probability of successful support recovery $\mathbb{P}(\hat{J} = J)$ as the number of times we obtain the exact support recovery among 100 repetitions, divided by 100. We compute the standard deviation as $(\mathbb{P}(\hat{J} = J)(1 - \mathbb{P}(\hat{J} = J)))/100$. Performance is measured by the empirical probability of correct classification $\mathbb{P}(\hat{J} = J)$ and maximal estimation error $\|\hat{\Pi} - \Pi\|_{\infty,\infty}$. To create error bars we use a empirical 95% confidence interval around the estimate.

**Uniform Distribution.** For all the experiments in this sub-section, we let $p = 80$, $k = 6$ and $|J| = 6$ and perform 100 repetitions of each setting. We now consider the setting of different dimensions. We choose $n \in \{3, 5, 7, 9\}$ and use $\rho = \sqrt{\frac{\log(p+1)}{mn}}$ for all pairs of $(m, n)$. $T$ is defined as above. For each $j \in [n]$, $X_j^{(i)} \sim 0.5\mathcal{N}(0, \Sigma^{(i)}) + 0.5\Delta_i$, where $\Delta_i$ is a $p \times 1$ vector of independent $Uniform(0,1)$ random variables. Figure 2 depicts the outcome of our experiments. For different choices of $n$, the graphs overlap each other perfectly (both $\mathbb{P}(\hat{J} = J)$ and $\left\|\hat{\Pi} - \Pi\right\|_{\infty,\infty}$). Then, we took the recovered support and derived the probability of support recovery for a novel task. We perform the experiments when there are $2, 3$, or $4$ extra zeros and in all of those experiments we have found greater than 95% probability of accurately identifying the extra zeros.

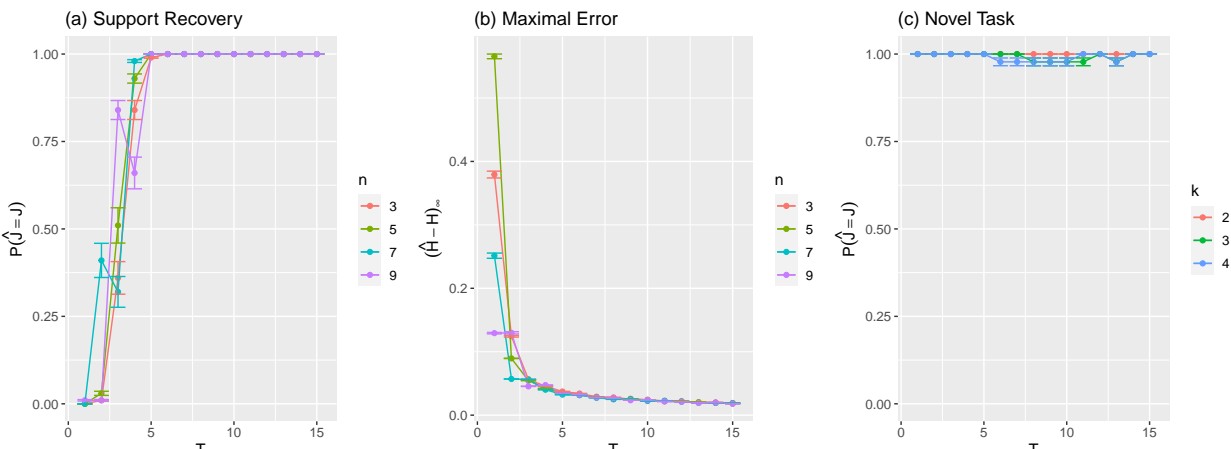

Figure 2: (a), (b): Simulations for Theorem 3.2 on the probability of support union recovery and maximal error under various settings of $n$. Let $\rho = \sqrt{\frac{\log(p+1)}{mn}}$. The x-axis is set by $T := \frac{mn}{\log(p+1)}$ varying from $\{1, \ldots, 15\}$ for support recovery. (c): Probability of support recovery for novel task. We let $\rho = \sqrt{\frac{\log(|J|+1)}{n}}$. The x-axis is set by $T := \frac{n}{\log(|J|+1)}$ varying from $\{10, \ldots, 25\}$

**Sub-Gaussian Distribution.** For all the experiments in this sub-section, we let $n = 5$, $k = 6$ and $|J| = 6$ and perform 100 repetitions of each setting. We now consider the setting of different dimensions. We choose $p \in \{40, 45, 50, 55\}$. $T$ and $\rho$ are defined as above. For each $j \in [n]$, $X_j^{(i)} \sim 0.5\mathcal{N}(0, \Sigma^{(i)}) + 0.5\Delta_i$, where $\Delta_i$ is a $p \times 1$ vector of independent $Exponential(0, 1)$ random variables. Figure 3 depicts the outcome of our experiments. For different choices of $p$, the graphs overlap each other perfectly (both $\mathbb{P}(\hat{J} = J)$ and $\left\|\hat{\Pi} - \Pi\right\|_{\infty,\infty}$). Then, we took the recovered support union and derived the probability of support recovery for a novel task. We perform the experiments when there are $2, 3$, or $4$ extra zeros and in all of those experiments we have found greater than 90% probability of accurately identifying the extra zeros.

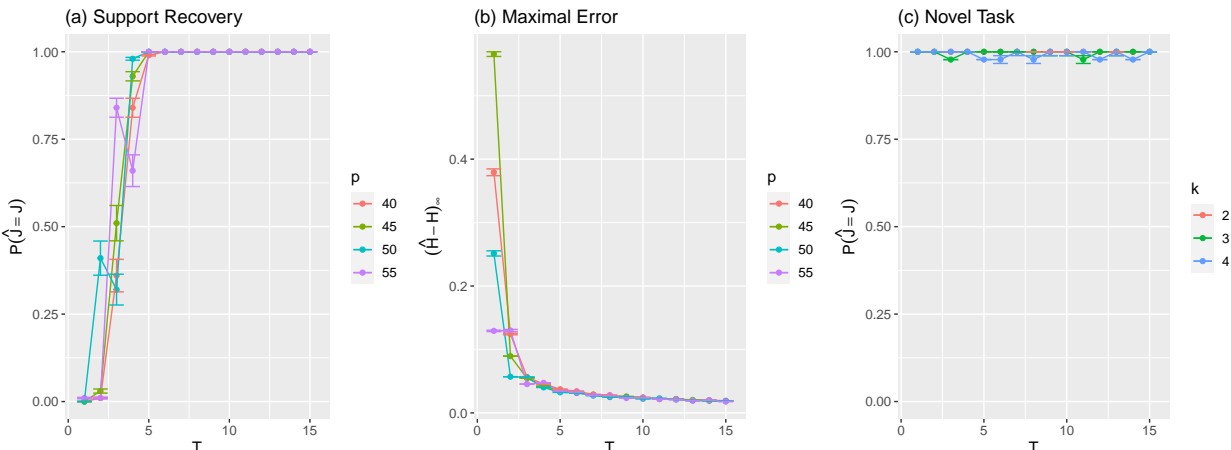

Figure 3: (a), (b): Simulations for Theorem 3.2 on the probability of support union recovery and maximal error under various settings of $p$. We let $\rho = \sqrt{\frac{\log(p+1)}{mn}}$. The x-axis is set by $T := \frac{mn}{\log(p+1)}$ varying from $\{1, \ldots, 15\}$ for support union recovery. (c): Probability of support recovery for novel task. We take $\rho = \sqrt{\frac{\log(|J|+1)}{n}}$. The x-axis is set by $T := \frac{n}{\log(|J|+1)}$ varying from $\{10, \ldots, 25\}$

## C.2 Comparison With Multi-Task Sparse PCA

In this section, we compare the meta-learning method against a multi-task learning method when applied to support union recovery under a Gaussian distribution setting. The method of generating the dataset and meta-learning is as described in Section 5. As before, we let $p = 30$, $|J| = 5$, and fix $n = 3$. The values we choose for $m$ are chosen from the set $\mathcal{M} \in \{2, 3, 4, 5, 6, 7, 9, 10\}$. We perform 24 replications and report the mean probability and the 95% confidence interval as the error bars around the mean. The results are plotted in Figure 4 (a). The penalty $\rho$ is always applied to be $\sqrt{\log(p+1)/mn}$. As previously noted, as the number of tasks increase, meta-learning achieves a perfect support recovery, while multi-task learning gets progressively worse.

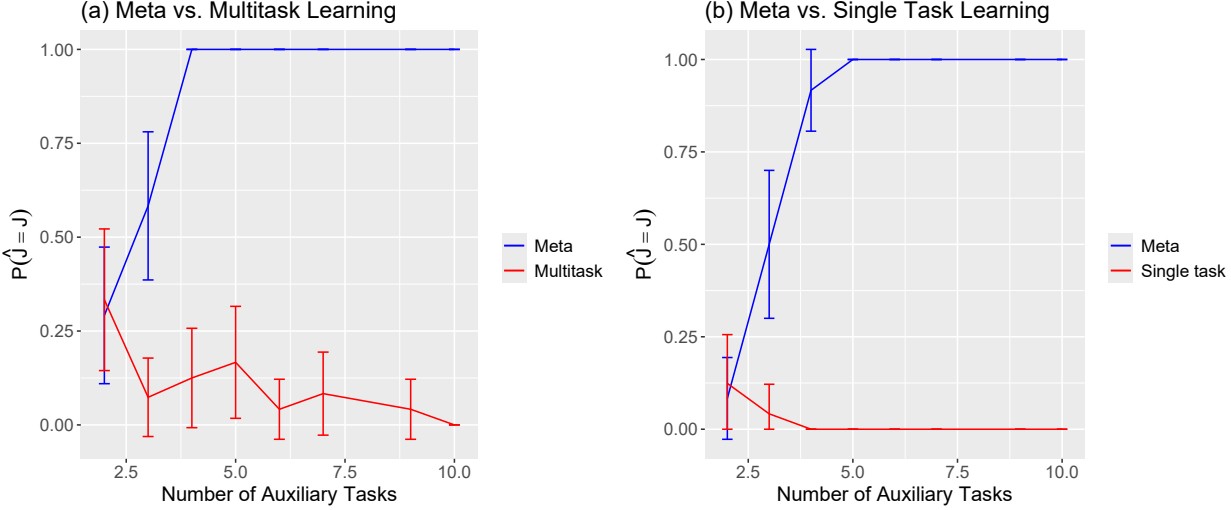

Figure 4: (a) Comparison showing the difference in the probability of recovering the correct support union of meta-learning versus multi-task learning. (b) Comparison showing the difference in the probability of recovering the correct support union of meta-learning versus single task sparse PCA. Y-axis : Probability of support union recovery. X-axis : Number of auxiliary tasks.

**Multi-Task Learning.** For the multi-task learning method, we maximise the following objective function:

$$\sum_{i=1}^{m} \langle S^{(i)}, H^{(i)} \rangle - \rho \| H^{(1)}, \ldots, H^{(m)} \|_1 \text{ subject to } H^{(i)} \in \mathcal{F}^k, \ \forall i \in [m],$$

where $S^{(i)}$ is the covariance matrix corresponding to task $i$ and $\| H^{(1)}, \ldots, H^{(m)} \|_1 = \sum_{j,k} \sup_{i \in [m]} |H_{j,k}^{(i)}|$. We apply $\sqrt{\frac{\log(p+1)}{mn}}$ as the penalty for the tasks. We recover the support union by taking the union of the recovered support.

**Meta-Learning.** For the meta-learning setup, we proceed according to Section 5 and recover the support union from the pooled covariance estimator. We apply $\sqrt{\frac{\log(p+1)}{mn}}$ as the penalty when $m$ is the total number of tasks.

As we can see, meta-learning outperforms multi-task learning with 100% accuracy for large values of $m$.

### C.3 Comparison with Single Task Sparse PCA

In this section, we compare the meta-learning method against a single task learning method when applied to support union recovery under a Gaussian distribution setting. The method of generating the dataset and meta-learning is as described in Section 5. We let $p = 30$, $|J| = 5$, and fix $n = 3$. The values we choose for $m$ are chosen from the set $\mathcal{M} \in \{2, 3, 4, 5, 6, 7, 9, 10\}$. We perform 24 replications and report the mean probability and the 95% confidence interval as the error bars around the mean. The results are plotted in Figure 4 (b).

**Single Task Learning.** For the single task learning setup there are $m$ many objective functions being maximised simultaneously. Let $S^{(i)}$ be the covariance matrix associated with the task $i$. Then the $i$-th objective is to maximise,

$$\langle S^{(i)}, H \rangle - \rho \| H \|_{1,1} \text{ subject to } H \in \mathcal{F}^k. \tag{C.1}$$

Let $\hat{H}^{(i)}$ be the maximiser of this objective function. Let $\hat{J}^{(i)} = supp(H^{(i)})$ and take $\hat{J} = \bigcup_{i \in \mathcal{M}} \hat{J}^{(i)}$ to be the natural estimator for the support union. We apply $\sqrt{\frac{\log(p+1)}{n}}$ as the penalty for each of the $m$ tasks.

**Meta-Learning.** For the meta-learning setup, we proceed according to Section 5 and recover the support union from the pooled covariance estimator. We apply $\sqrt{\frac{\log(p+1)}{mn}}$ as the penalty when $m$ is the total number of tasks.

Single task learning of sparse PCA can be seen to be completely useless whenever the number of tasks get large and fails to recover the correct support union after a while. On the other hand, meta-learning improves steadily with the increasing number of tasks and the results coincide with our experiments in Section 5.

### C.4 Real World Datasets

**Brain Imaging Data.** The 1000 functional connectomes dataset is from nitrc.org and contains resting-state functional magnetic resonance signals from 1128 subjects, 41 sites around the world, and 157 brain regions. This dataset consists of labs around the world that collected MRI from different subsets of subjects/persons. Each task is a different research lab with different magnetic resonance imaging scanner/equipment with different physical properties, and each brain region forms a covariate. Exploratory research like PCA is often performed to unveil important brain regions, which is the support. Although different labs might not have the same important brain regions (support) it is reasonable to believe that there are some similarities across labs. This dataset is therefore a prime candidate for meta-learning. We use a set of independent samples to construct the true covariance matrix, and thus the true support union. We then constructed the pooled

estimator using $m$ auxiliary tasks, where $m \in \{1, 2, 3, 5, 8, 10, 12, 15, 18, 20, 25, 30, 35, 41\}$. Then we compared the recovered support with the true support to calculate the mean percentage and standard deviation of error in the recovered support. The error bars were obtained by creating a 95% confidence interval around the mean. As we clearly see in Figure 5 the probability of incurring an error decreases as we increase the number of tasks.

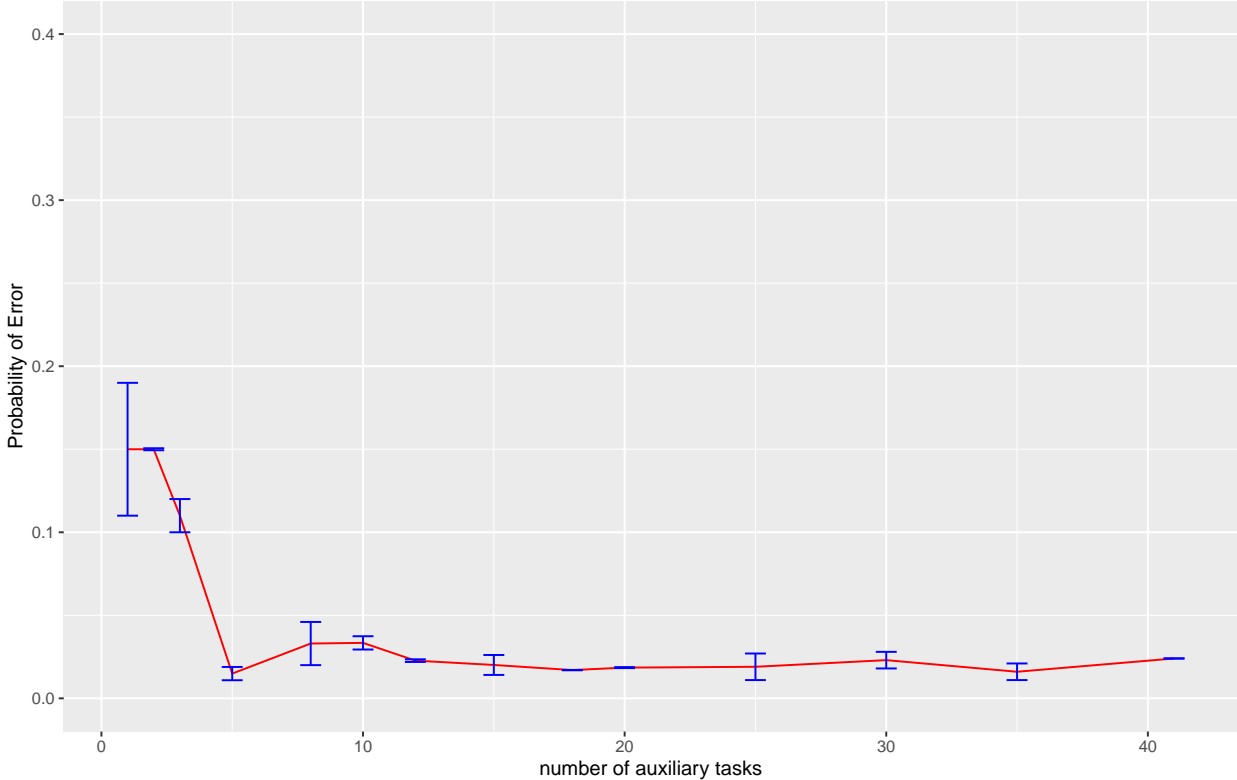

Figure 5: Probability of error of support recovery for brain imaging dataset. We took an increasing number of auxiliary tasks $m \in \{1, 2, 3, 5, 8, \ldots, 30, 35, 41\}$. The probability of error decreases.

**Recovered Broadmann areas in the support union.** At a penalty of $\rho = 10\sqrt{\frac{\log(p+1)}{mn}}$, the recovered significant brain regions were: Broadmann areas 6, 17, and 44. Interestingly, these are widely reported in other fMRI studies, as shown in Table 2.

| Area | Found significant in |
|------|----------------------|
| 6 | Rubia et al. (2000); Shuster & Lemieux (2005); Sylvester et al. (2003); Fincham et al. (2002) |
| 17 | Yan et al. (2005); Fulford et al. (2018); Wohlschläger et al. (2005); Tan et al. (2001) |
| 44 | Baron-Cohen et al. (2006); Friederici et al. (2000); Fincham et al. (2002); Binkofski et al. (1999) |

Table 2: Recovered Broadmann areas in the support union

**Cancer Genetics Data.** The cancer genome atlas program dataset is available at gdc.cancer.gov and contains the genetic sequencing of 1576 patients. In cancer research, exploratory research is often performed to discover the important genes, which forms the support of PCA studies. In this dataset, we have 11 types of cancer, each one being an auxiliary task, and each gene is a covariate. Different cancer types might not elicit the same important genes (support) but it is reasonable to believe that there are some similarities across cancer types. Therefore, this dataset is a prime candidate for meta-learning. We use a set of independent samples to construct the true covariance matrix and the true support union. We then constructed the pooled

estimator using $m$ auxiliary tasks where $m \in \{1, \ldots 10\}$. Then we compared the recovered support with the true support to calculate the mean percentage and the standard deviation of the error in recovered support. The error bars were obtained by empirically calculating the 95% confidence interval around the mean. As we can see in Figure 6 the probability of incurring an error decreases as we increase the number of tasks.

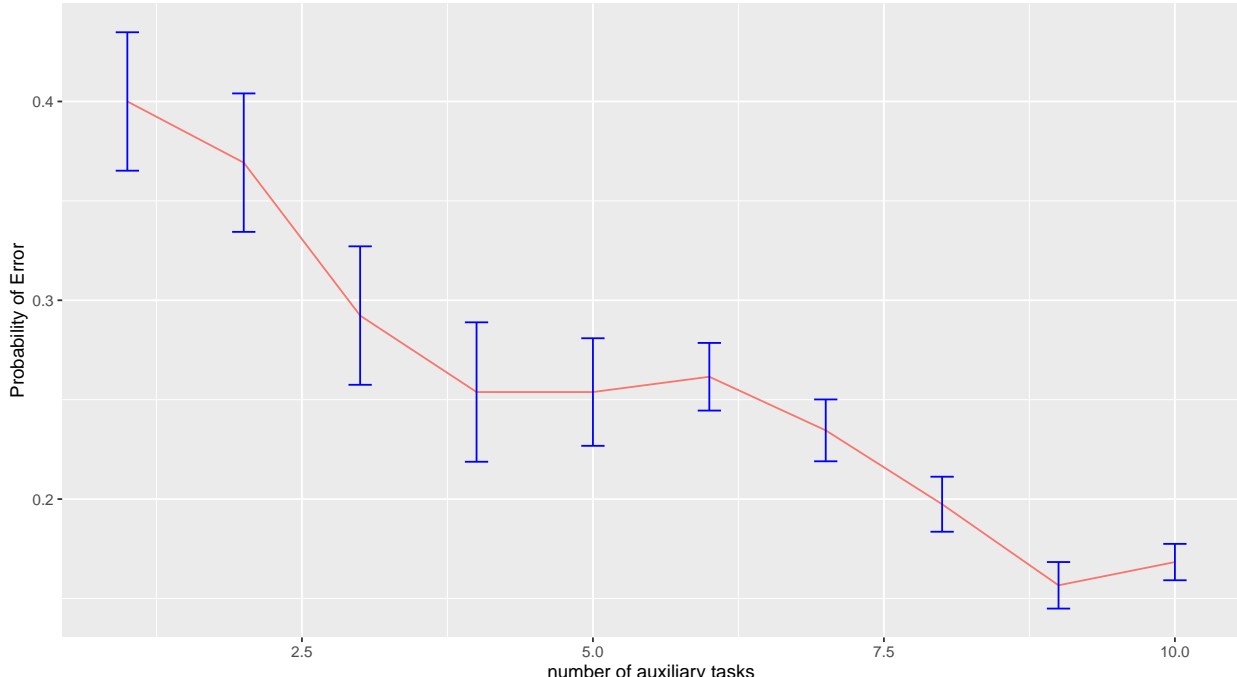

Figure 6: Probability of error of support recovery for cancer genetics dataset. We took an increasing number of auxiliary tasks $m \in \{1, 2, \ldots, 10\}$. The probability of error can be seen to be decreasing.

**Recovered genes in the support union.** At a penalty of $\rho = 10\sqrt{\frac{\log(p+1)}{mn}}$, the recovered significant genes were: PTP4A3, CLEC3B, E2F3, PSMA7. Interestingly, these genes are widely reported in other studies of cancer research, as shown in Table 3.

| Gene | Found significant in |
|---|---|
| PTP4A3 | Huang et al. (2014); Den Hollander et al. (2016); Wang et al. (2019); Zimmerman et al. (2013) |
| CLEC3B | Sun et al. (2020); Zhu et al. (2019); Dai et al. (2019); Zhang et al. (2017) |
| E2F3 | Feber et al. (2004); Cooper et al. (2006); Gao et al. (2017); Olsson et al. (2007) |
| PSMA7 | Hu et al. (2009; 2008b;a); Du et al. (2009) |

Table 3: Recovered genes in the support union

