# OpenReview forum: "Meta Sparse Principal Component Analysis"
_TMLR — Withdrawn by Authors_

### Review · Reviewer_ZCie · 2024-12-06

**Summary Of Contributions:**

This paper is concerned with the meta learning of a sparse principal component analysis. The setting is as follows. One has access to a few samples $n$ of many tasks $m$ and is interested in learning a single principal component (PC) matrix from all tasks such that it performs well for new tasks. In comparison, multi-task learning would learn a PC matrix for each task and, thus, would require a larger sample size per task. The authors derive (i) the optimal number $m$ of tasks to estimate the PC matrix, (ii) the optimal number of samples $n$ given a number of tasks $m$, and (iii) the optimal number of samples for a new task, given the PC matrix estimated from $m$ tasks. Their claims are supported by some synthetic experiments.

**Audience:**

Yes

**Claims And Evidence:**

No

**Requested Changes:**

1. Please provide answers to my questions.
2. Please respond to the weaknesses raised.
3. Fix the notation and typos.
4. Include convincing real-world experiments in the main text.
5. Please make clear what the points of the experiments are.

**Strengths And Weaknesses:**

### Strengths
1. Principal component analysis (PCA) and also its sparse version are important techniques in machine learning. This paper contributes to a better understanding of the meta learning aspect of sparse PCA.
2. Mathematical analyses and guarantees on the number of tasks and sample sizes are derived and they *appear* to be okay. (I did not check the appendix with thirteen pages of proofs in time.)
3. The paper is mostly well written and easy to follow.


### Weaknesses
1. High-dimensionality is mentioned in the abstract and introduction but no high-dimensional scenario is considered. Information on the dimensionality of the real-world data is missing.
2. As of now, I do not see a convincing use case of this paper. See my second question for more details.
3. The experimental evaluation is lacking and details are missing. See my third question for more details.
4. The clarity should be improved. Some variables have to be guessed from context (e.g., $n^{(i)}$, $J^c$), others remain unclear (e.g., $\alpha$).
5. Interesting parts are deferred to the appendix although there is space in the main paper.


### Questions
1. If multi-task learning and meta learning are two different things, how can one be superior to another? (See Page 2 before Section 1.1 and Page 10 on the bottom.) Isn't it like claiming that classification is better than regression for a classification task? Note that the experiment supporting this claim is also not present in the main text but deferred to the appendix. I think it should be noted that this statement is limited to the low sample size scenario, because if I have a lot of samples per task, why should meta-learning be better?
2. Can you provide some convincing use cases of this paper? Where am I interested in computing
	- a single sparse and high-dimensional PCA
	- from a large number of tasks $m$
	- where only a few samples $n$ per task are available
	- where the true PC matrices per task only differ by being random pertubations of the true underlying PC matrix
	- where each observation is distributed according to a sub-Gaussian?
	Take, for example, the brain image data set from Appendix C.4. Why should that be sub-Gaussian and why should the tasks all be pertubations of a single underlying PC matrix?
3. What are the details of the experiments. How where they optimized? Why are the number of repetitions different across experiments? Why are no more scenarios evaluated ($p,k,|J|,m,n$)? What happens if I deviate from the optimal scenario? How can I interpret $T$ in the experiments? How do the matrices $\Sigma^{(i)}$ differ? WHat is the distribution?
4. Couldn't I forget about the tasks and treat all samples from all tasks as samples and estimate one sparse PCA from it and compare against it as a baseline?



### Minor Comments
- Abstract: If brain and cancer genetics data sets are mentioned so prominently in the abstract, they should appear in the main paper, not only in the appendix.
- Page 1 Paragraph 2: "can exhibit unexpected behavior" is not really "particular".
- Page 2 Section 1.1: Commas missing in $\Sigma^{(1)},\ldots,\Sigma^{(m)}$ and $\Pi^{(1)},\ldots,\Pi^{(m)}$.
- Page 2 Section 1.1: Here, the reader gets the impression that $J$ carries tuples, i.e., $(i,j)$ positions in a matrix but later it seems one only cares about the diagonal elements. Please be more specific. The support in Table 1 is also related.
- Table 1: $\Sigma$ The true covariance **of** $A$ (of is missing)
- Definition 2.1: The symbol $n^{(i)}$ was never introduced.
- Page 4 Paragraph 1: Tense mismatch, *"we focus"* instead of *"we focused"*.
- Between Equations (2.1) and (2.2): Shouldn't it be $\lambda_l$?
- Section 2.2 second sentence: I cannot recall what I didn't read. Improper wording.
- Section 2.2: The symbol $H$ could have been introduced before using it.
- Section 2.2: The notation was already abused between Equations (2.1) and (2.2).
- Remark 2.3: Mistake in "$\Pi$ is 0 is non-zero if".
- Before Equation (2.7): **Equation** (2.6)
- Equation (2.7): Why $\hat{\pi}$ instead of $\hat{\Pi}$?
- Lemma 3.1: The summation is missing its limits.
- Before Assumption 4: There is no Assumption 3.2. It is Assumption 3.
- Assumption 4: What is $J^c$ and what is $\alpha$?
- Theorem 3.3: .,
- Various places (e.g., Sec. 3.1): Math is always part of a sentence, not standing alone.
- Figure 1: (c) does not show T from $\{10,\ldots,25\}$ as claimed in the caption.
- Page 9: The page limit for a non-long TMLR paper is 12 pages. This paper has slightly more than 10. I wonder why the real world experiments are not presented in the main body of the paper. There is space. Those results are also mentioned in the abstract.
- Page 11: Expand names in Bai et al. (2007) as everywhere else. Same with Olsson. Steven Boyd appears twice in the same paper.
- Page 31: Tense mismatch in "we use" and "we then constructed".
- Figures 5 and 6: Unnecessarily large.

---

### Review · Reviewer_5Vav · 2024-12-07

**Summary Of Contributions:**

The paper studies meta learning the support of sparse principal component analysis. The general setup is that $\Sigma$ is a covariance matrix where the first $k$ eigenvectors are sparse. Then they consider multi-tasks obtained by considering random perturbations $\Sigma^{(i)}$ of $\Sigma$. They show optimal sample complexity of support recovery through the pooled covariance estimator (under suitable assumptions on the randomness). Moreover, they show that novel tasks can be adapted to with $O(1)$ samples. Finally, empirical support on brain data is provided.

**Audience:**

Yes

**Claims And Evidence:**

No

**Requested Changes:**

Mostly questions and remarks:

- In the abstract "where a task is defined as a random Principal Component (PC) matrix with its own support." This is unclear (at least at this point).
- The preliminaries Section (except for the table) up to Section 2.1 was not very helpful and this discussion could be mostly postponed in my opinion.
- What is the motivation for 2.3? It should be made clear that probably $R$ is close to the identity and $D$ close to 0.
- The equation between 2.4 and 2.5 can probably be removed (or do we need to consider random $H$)?
- Lemma 3.1 essentially seems to be a coordinate-wise standard union bound (which do not require identical distributions), this (or my oversight) could be clarified.
- I have difficulties to understand Assumption 3: What are examples for 3.1? Doesn't this entail that $\mathbb{E}(R_{ij}^2)=0$ for $i\neq j$ (and $\mathbb{E}(R_{ii}R_{jj})=1$ which implies $R_{ii}=R_{jj}$?
- Regarding Assumption 3.3: If $L$ is large the order of the eigenvalues can be permuted which raises questions regarding Theorem 4.2.
- In Theorem 3.2: What does $C_{\pi}$ depend on (also the font of the constants confused me (maybe use standard font or a frak font).
- In the current form Theorem 3.2 is hard to parse. Maybe express this as requirements on the sample size and number of tasks, also provide the scaling in the most important problem parameters (probably $J$). Restating the condition does not provide much additional insight.
- In Theorem 3.3: Which assumptions of Theorem 3.2 exactly do you assume here? Do we assume the bounds hold?
- Equations 3.9-3.11 are unclear and probably refer to the proof in prior work (maybe move to the appendix or expand).
- I could not follow Section 4 and this is related to the assumptions on the perturbations. On an intuitive level: If $R^i$ is not the identity it can change the support of the eigenvectors substantially and if it is not a rotation it can also change the spectrum. Also, the assumptions on $D$ seem not to imply that the first $k$ eigenvalues are preserved so that the support of the projection can change substantially. On a formal level this appears, e.g., in the second line of the proof of Proposition 4.1 which states 'We also know from the recovered support...' which I cannot relate to the support of the new task.
- Regarding Theorem 4.2 a similar comment as for Theorem 3.2 applies: In the current form it is hard to understand the main message.

**Strengths And Weaknesses:**

The paper studies a problem that is generally of interest and it provides rigorous and tight results.
While the proof techniques seem to be mostly standard, the proofs are still quite technical.

On the negative side, I found this paper quite hard to follow (I cannot really pinpoint this but see my questions below). It would be good if additional intuition for the scaling of the results and the relevance of the proposed setting can be provided.

---

### Review · Reviewer_46Z2 · 2025-01-22

**Summary Of Contributions:**

The paper analyses a meta learning algorithm for estimating the support of the principal components in high dimensional PCA. This algorithm relies on sparse PCA using a regularization term based on the $||\cdot||_{1,1}$ norm.

The contributions are two-fold:

- They exhibit minimax sample complexity for estimating this support in terms of the number of samples per task $n$, the number of tasks $m$, and the size of the support $J$. This complexity is given by $n = \mathcal{O}(|J|\sqrt{m^{-1} \log{p}})$.

- Given that the support $J$ has been correctly estimated, they show that the optimal sample complexity for estimating a novel task is $n = \mathcal{O}(\log{|J|})$

The results of the paper rely on a sub-gaussian assumption on the data that has already been explored in the meta-learning literature.

The proposed statistical setup constitutes of an underlying task parameterized by a main covariance matrix and of a family of related tasks governed by covariances matrices which are noisy versions of the main one. Suitable assumptions on the noise allow to show that there exists a range of hyperparameter values for the sparsity encoding regularization parameter such that with high probability, the support of the underlying task is recovered (Theorem 3.2).

Once the underlying support is learned, one can leverage this information when learning a new task by parameterizing the support of this new task on the underlying support, resulting in reducing the parameter size to estimate. This strategy is shown to be working with high probability in Theorem 4.2.

**Audience:**

Yes

**Broader Impact Concerns:**

Does not apply here.

**Claims And Evidence:**

No

**Requested Changes:**

- [Major] Fix notation and typos
- [Major] Improve clarity and exposition of references
- [Minor] Plots should be remade as they leave an unprofessional feeling. Check the fonts therein. Use a colormap that makes sense for an increase in the number $n$.

**Strengths And Weaknesses:**

Strenghts:

- The paper is of interest to the TMLR community
- The paper seems sound and is mostly well-written, though dense at times.
- The results are novel and precious.
- The statistical assumptions are reasonnable.

Weaknesses:

There are to me two major weaknesses here, related to references and clarity of the results.

- Problem 2.7 has been known in the literature (at least) as Fantope projection and selection. It would be nice to have a bit of context on this algorithm specifically at the point of its introduction. Relevant work [e.g. Vu et al. 2013] is cited elsewhere in the paper but also belongs here in my opinion. It would be the occasion to introduce challenges associated with this estimator (unicity of the solution for example that makes a bridge to Assumption 1).
- You mention that learning a new task has sample complexity $\mathcal{O}(\log(p))$. From what I understood, this comes from [Deng, 2019] where the multi-task PCA is investigated, with sample complexity $n = \mathcal{O}(m \log(mp))$. Is there a better reference to these results than a PhD thesis, in the form of peer-reviewed conference or journal papers ? It is unclear to me how Corollary II.5 implies the claim given the difference in setups, and the reader should not have to invest so much time in digesting the reference.


- Theorem 3.2 is tough to interpret. In particular, the condition that has to be satisfied by the regularization parameter $\rho$ in order to get the theoretical guarantees is hard to grasp. It involves many quantities depending on the hypotheses and there is little explanation about the rate at which these quantities shrink when $n, m, p$ are getting larger.
- The same can be said about Theorem 4.2. I struggle to see how it implies the sample complexity $n = \mathcal{O}(\log(|J|)$.

Notation/Typos:

- [Section 2.2] Clarify the notation used when writing $\mathrm{Cov}(X, HX | X)$.
- [Assumption 1] It was not stated before that $k$ is fixed.
- Notation $\Sigma_{J^c, J}$ is used but never defined.
- [Theorem 3.2] Typo below the theorem $\lambda^{\dagger} = (\lambda_1(\Sigma) + ...$

Further questions/remarks:

- The role of $k$ is seldom mentioned. It is left fixed in the experiments. I would have been interested in knowing the effect of $k$ on the solution depending on the other hyperparameters.
- How interesting is it in practice to recover the support of $\Pi$, as opposed to having information about the supspace spanned by the principal components associated to the underlying and individual tasks ?
- Could you elaborate on the differences it makes to work on the rank-k Fantope $\mathcal{F}$ rather than on the Stiefel manifold, which is parameterized by lower-dimensional matrices $H \in \mathbb{R}^{p \times k}$ ?
- In practice, you used $\rho = \sqrt{\frac{\log(p+1)}{mn}}$. How does this relate to theorem 3.2 ? Especially on the upper bound on $\rho$.

---

### Note · Authors · 2025-02-06

**Comment:**

Dear Action Editor and reviewers, we sincerely thank you for your very thoughtful feedback and we already implemented several changes in our manuscript. While working on those, we found that the assumptions and the theory would require a major revision. Thus, we decided to withdraw our manuscript and resubmit it later. We deeply regret our mistake, and feel very sorry for the inconvenience.

**Withdrawal Confirmation:**

I have read and agree with the venue's withdrawal policy on behalf of myself and my co-authors.